# Symmetry-enforced minimal entanglement and correlation in quantum spin chains

Kangle Li[1] and Liujun Zou[2, *]

[1]*Department of Physics, Hong Kong University of Science and Technology, Clear Water Bay, Hong Kong SAR, China*
[2]*Department of Physics, National University of Singapore, Singapore 117542*

The interplay between symmetry, entanglement and correlation is an interesting and important topic in quantum many-body physics. Within the framework of matrix product states, in this paper we study the minimal entanglement and correlation enforced by the $SO(3)$ spin rotation symmetry and lattice translation symmetry in a quantum spin-$J$ chain, with $J$ a positive integer. When neither symmetry is spontaneously broken, for a sufficiently long segment in a sufficiently large closed chain, we find that the minimal Rényi-$\alpha$ entropy compatible with these symmetries is $\min\{-\frac{2}{\alpha-1}\ln(\frac{1}{2^\alpha}(1+\frac{1}{(2J+1)^{\alpha-1}})), 2\ln(J+1)\}$, for any $\alpha \in \mathbb{R}^+$. In an infinitely long open chain with such symmetries, for any $\alpha \in \mathbb{R}^+$ the minimal Rényi-$\alpha$ entropy of half of the system is $\min\{-\frac{1}{\alpha-1}\ln(\frac{1}{2^\alpha}(1+\frac{1}{(2J+1)^{\alpha-1}})), \ln(J+1)\}$. When $\alpha \to 1$, these lower bounds give the symmetry-enforced minimal von Neumann entropies in these setups. Moreover, we show that no state in a quantum spin-$J$ chain with these symmetries can have a vanishing correlation length. Interestingly, the states with the minimal entanglement may not be a state with the minimal correlation length.

## I. Introduction

Symmetry, entanglement and correlation are three important concepts in quantum physics. It is well known that certain symmetries can force the system to possess special patterns of entanglement and correlation. The most familiar and elementary example is that an $SO(3)$ spin rotation symmetry forces a pure state of two qubits to be maximally entangled and be a spin singlet. The many-body generalizations of such symmetry-enforced entanglement and correlation are much richer and more nontrivial, and we list a few examples for illustration. 1) Spontaneous symmetry breaking in the ground states of a quantum many-body system results in the Greenberger–Horne–Zeilinger-type (GHZ) long-range entanglement and long-range correlation [1–3]. 2) In the entanglement-enabled symmetry-breaking orders introduced in Ref. [4], there must be some other nontrivial structures of entanglement, in addition to the GHZ entanglement that

* lzou@nus.edu.sg

is common in all spontaneous symmetry breaking orders. 3) The Lieb-Schultz-Mattis-type (LSM) theorems dictate that certain symmetry conditions can force a many-body system to possess long-range entanglement and correlation, even if the symmetry is not spontaneously broken [5–11]. 4) In mixed many-body states, some symmetries can also impose nontrivial patterns of entanglement [12–14].

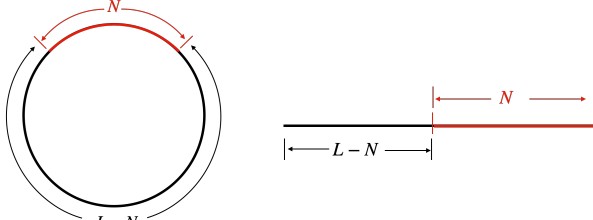

FIG. 1. In a chain with $L$ sites, we are interested in the von Neumann and Rényi entropies of a subsystem with $N$ contiguous sites (colored in red), where $1 \ll N \ll L$. The left is a closed chain with two entanglement cuts, and the right is an open chain with one entanglement cut. The closed chain is naturally compatible with a lattice translation symmetry, and the open chain can be translation symmetric when $L \to \infty$. For any $\alpha \in \mathbb{R}^+$, we denote the Rényi-$\alpha$ entropy in these two cases by $S_\alpha(\rho_{\text{t.c.}}(N,L))$ and $S_\alpha(\rho_{\text{o.c.}}(N,L))$, respectively, with $\rho_{\text{t.c.}}(N,L)$ ($\rho_{\text{o.c.}}(N,L)$) being the reduced density matrix of the subsystem, where "t.c." ("o.c.") stands for two cuts (one cut). For the special case with $\alpha = 1$, these Rényi entropies become von Neumann entropies, simply denoted by $S(\rho_{\text{t.c.}}(N,L))$ and $S(\rho_{\text{o.c.}}(N,L))$, respectively.

Despite these previous studies, some basic questions regarding the interplay between symmetry, entanglement and correlation remain unanswered. For example, if a system enjoys certain symmetries, what is the *minimal* entanglement of this system, quantified by entanglement measures such as the entanglement entropy of a large subsystem (see Fig. 1)? The minimally entangled states should obey the entanglement area law [15], which, in a one dimensional (1D) system, means that the von Neumann entropies $S(\rho_{\text{t.c.}}(N,L))$ and $S(\rho_{\text{o.c.}}(N,L))$ in Fig. 1 are finite as $N$ and $L - N$ go to infinity. Under the assumption that the limits $\lim_{(N,L-N)\to(\infty,\infty)} S_\alpha(\rho_{\text{t.c.}}(N,L))$ and $\lim_{(N,L-N)\to(\infty,\infty)} S_\alpha(\rho_{\text{o.c.}}(N,L))$ in Fig. 1 exist, which is natural for translation symmetric states with little entanglement, we would like to find out the smallest possible values of $\lim_{(N,L-N)\to(\infty,\infty)} S_\alpha(\rho_{\text{t.c.}}(N,L))$ and $\lim_{(N,L-N)\to(\infty,\infty)} S_\alpha(\rho_{\text{o.c.}}(N,L))$ compatible with the symmetries, for all $\alpha \in \mathbb{R}^+$.

Similarly, what is the *minimal* correlation length of this system due to these symmetries? In particular, can some symmetries force the system to have a nonzero correlation length? Is a state with the minimal entanglement entropy also a state with the minimal correla-

tion length? Note all these questions are about *states*[1], and, a priori, we do not have to refer to any Hamiltonian to discuss them.

In this paper, we address these questions in the context of quantum spin-$J$ chains with an $SO(3)$ spin rotation symmetry and a lattice translation symmetry that are not spontaneously broken, where $J \in \mathbb{Z}^+$ (see Fig. 2). Concretely, we consider a 1D spin system where each site is described by a $(2J+1)$-dimensional Hilbert space, and the total Hilbert space of the entire system is the tensor product of all the local Hilbert spaces. The degree of freedom at each site transforms as a spin-$J$ representation under $SO(3)$, and they are shifted from one site to the next under translation.

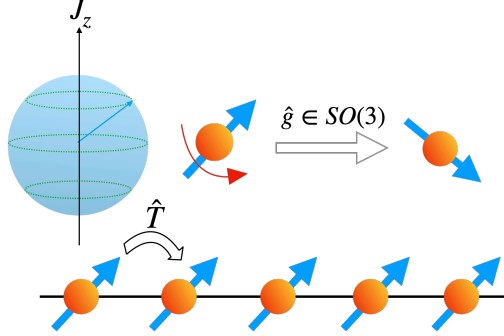

FIG. 2. Cartoon of a quantum spin chain with $SO(3)$ and translation symmetries. The blue sphere shows the discretized $J_z$ values. The action of an $SO(3)$ operation denoted by $\hat{g}$ and the action of translation $\hat{T}$ are sketched.

This setup is interesting for multiple reasons. First, the quantum spin problems are fundamental in both theoretical and experimental studies of quantum many-body systems, so it is useful to thoroughly understand them [16–18]. Second, the structures of entanglement and correlation in 1D systems are much better understood than their higher dimensional counterparts, so more concrete conclusions can be reached. For example, the ground states of a large class of gapped 1D Hamiltonians obey the entanglement area law [11, 19–23]. Meanwhile, for gapless 1D systems described by a conformal field theory at low energies, $S(\rho_{\text{t.c.}}(N,L)) = 2S(\rho_{\text{o.c.}}(N,L)) \sim \ln N$ when $1 \ll N \ll L$ [24–27]. Moreover, many 1D states can be efficiently represented by matrix product states (MPS) [28–34]. Third, the following observation suggests some interesting constraints on the entanglement and correlation from the symmetries in this setup. The LSM theorems imply that a quantum spin-$J$ chain with $SO(3)$ and translation symmetries must be long-range entangled if $J \in \mathbb{N} + \frac{1}{2}$. However, the case with $J \in \mathbb{Z}^+$ and

---

[1] In the rest of this paper, all states are assumed to be pure unless otherwise stated.

the case with $J \in \mathbb{N} + \frac{1}{2}$ are expected to share similar physical properties in the semi-classical limit where $J \gg 1$. So although a symmetric chain with $J \in \mathbb{Z}^+$ can satisfy the entanglement area law and have a finite correlation length, for it to behave similarly as a chain with $J \in \mathbb{N} + \frac{1}{2}$ when $J \gg 1$, the minimal von Neumann entropy of a long segment in a chain with $J \in \mathbb{Z}^+$ should diverge as $J$ increases, and the minimal correlation length of such a chain is also expected to diverge as $J \to \infty$.

Because in this paper we are after the states in such a quantum spin-$J$ chain that have the *minimal* entanglement and correlation length, it is natural to represent the states by translation invariant MPS, which are suitable for describing states with finite entanglement and correlation. For these states, the limits $\lim_{(N,L-N)\to(\infty,\infty)} S_\alpha(\rho_{\text{t.c.}}(N,L))$ and $\lim_{(N,L-N)\to(\infty,\infty)} S_\alpha(\rho_{\text{o.c.}}(N,L))$ indeed exist for all $\alpha \in \mathbb{R}^+$. From now on we will denote these limits as $S_\alpha(\rho_{\text{t.c.}})$ and $S_\alpha(\rho_{\text{o.c.}})$ for simplicity. Within this setup, we establish the following results.

- In any state of a quantum spin-$J$ chain ($J \in \mathbb{Z}^+$) with $SO(3)$ spin rotation symmetry and lattice translation symmetry that are not spontaneously broken, $S_\alpha(\rho_{\text{t.c.}}) \geqslant 2\min\{\ln(J + 1), -\frac{1}{\alpha-1}\ln(\frac{1}{2^\alpha}(1 + \frac{1}{(2J+1)^{\alpha-1}}))\}$ and $S_\alpha(\rho_{\text{o.c.}}) \geqslant \min\{\ln(J + 1), -\frac{1}{\alpha-1}\ln(\frac{1}{2^\alpha}(1 + \frac{1}{(2J+1)^{\alpha-1}}))\}$. The special case with $\alpha = 1$ gives the symmetry-enforced lower bounds of the von Neumann entropies, i.e., $S(\rho_{\text{t.c.}}) \geqslant \min\{2\ln(J + 1), \ln 4(2J + 1)\}$ and $S_\alpha(\rho_{\text{o.c.}}) \geqslant \min\{\ln(J + 1), \ln 2\sqrt{2J+1}\}$, which indeed diverge as $J \to \infty$.

- The simplest minimally entangled states saturating the above lower bounds take particular forms, which are referred to as type-I and type-II states in Eq. (20). For given values of $J$ and $\alpha$, whether the type-I or type-II state saturates the minimal entropy is shown in Fig. 6. For each such minimally entangled state, there exists a gapped local Hamiltonian with $SO(3)$ and translation symmetries, whose unique ground state is this state. In particular, in a spin-1 chain the Affleck-Kennedy-Lieb-Tasaki (AKLT) state [35, 36] is a minimally entangled state, which saturates the lower bound of the Rényi-$\alpha$ entropy for all $\alpha > 0$.

- No state of a quantum spin-$J$ chain with $SO(3)$ and translation symmetries can have a vanishing correlation length. However, calculating the minimal correlation length or proving that the minimal correlation length diverges as $J$ increases is beyond the scope of this work.

- A state with the minimal entanglement does not have to be a state with the minimal correlation length. In particular, the AKLT state does not have the minimal correlation length.

The rest of the paper is organized as follows. In Sec. II we review some basic facts about MPS, focusing on the aspects of entanglement and correlation length. In Sec. III we discuss the minimal entanglement of symmetric uniform MPS. We first present the main theorem on the symmetry-enforced minimal von Neumann entanglement entropy and then take three steps to prove it from Sec. III A to Sec. III C. In Sec. III D, we construct some minimally entangled states explicitly and verify that they saturate the lower bounds of the entanglement entropies. We then discuss the minimal Rényi-$\alpha$ entropy with a general $\alpha \in \mathbb{R}^+$ in Sec. III E. In Sec. IV, we discuss the symmetry-enforced correlation length from two different perspectives. Finally, we finish this paper by discussing our working assumptions and some open problems in Sec. V. Various appendices contain additional technical details, some of which may be of independent interest. For example, in Appendix B we discuss how an infinite chain state as a limit of some finite open chain states can have $SO(3)$ and translation symmetries, and in Appendix H we present results regarding the symmetry-enforced minimal entanglement entropies for all $SO(3)$-symmetric translation invariant MPS, which may spontaneously break the symmetry.

## II.  Basics of MPS

In this section, we review some basics of uniform MPS (uMPS). We first introduce the definition of uMPS and the left-canonical and right-canonical forms, along with the concept of transfer matrices. Next, we discuss the properties of eigenvalues and eigenvectors of transfer matrices, which determine the entanglement and correlation of the state. In particular, we distinguish two classes of states, the "injective uMPS" and "non-injective uMPS", which display crucially different properties in eigenvalues and eigenvectors. Lastly, we introduce the constraints on the uMPS structures due to the $SO(3)$-symmetry, which are essential for the subsequent sections.

### A.  MPS and its canonical forms

We consider a quantum state in the Hilbert space $\mathcal{H} = (\mathbb{C}^d)^{\otimes L}$, characterizing a system in one spatial dimension with $L$ sites, where each site is described by a $d$-dimensional local Hilbert space. A rank-3 tensor $A^i_{\alpha_1,\alpha_2}$ is formed by $d$ matrices $A^i$ for $1 \leqslant i \leqslant d$, with each $A^i$ being a $D \times D$ matrix. The $D$-dimensional vector space associated with $A^i$ is called the bond Hilbert space. A uniform matrix-product state (uMPS) is defined as

$$|\psi[A]\rangle = \sum_{\{i_x\}} \text{Tr}\left[A^{i_1} A^{i_2} \cdots A^{i_L}\right] |i_1 i_2 \cdots i_L\rangle. \quad (1)$$

This is a translation-invariant state on a closed chain (see Fig. 1).

Given a uMPS tensor $A^i$, if it satisfies $\sum_i A^i (A^i)^\dagger = \mathbb{1}$, then $A^i$ is said to be *right-canonical*. If it satisfies $\sum_i (A^i)^\dagger A^i = \mathbb{1}$, it is said to be *left-canonical*.

A uMPS can also be defined on an open chain with $L$ sites,

$$|\psi[A], a_l, a_r\rangle = \sum_{\{i_x\}} (a_l^{i_1})^\mathbf{t} A^{i_2} \cdots A_{i_{L-1}} a_r^{i_L} |i_1 i_2 \cdots i_L\rangle \tag{2}$$

where $a_l^i$ ($a_r^i$) are a set of $D$-dimensional vectors. In general, the $SO(3)$ and translation symmetries are explicitly broken in this state if $L$ is finite, but they are not explicitly broken in the thermodynamic limit where $L \to \infty$, if appropriate choices of $a_l^{i_1}$ and $a_r^{i_L}$ are made. The precise conditions of $a_l^{i_1}$ and $a_r^{i_L}$ under which the state is symmetric when $L \to \infty$ are discussed in Appendix B. Below we always assume that such choices are made. Note although these symmetries are not explicitly broken with such choices, they may still be spontaneously broken, if no other constraint on the uMPS is imposed[2].

## B. Entanglement and correlation from the transfer matrix

To study the entanglement and correlation in a uMPS, it is useful to consider the transfer matrix. The transfer matrix associated with a uMPS tensor $A^i$ is a $D^2 \times D^2$ matrix defined as $T[A] = \sum_i (A^i)^* \otimes A^i$, or equivalently,

$$T[A]_{(\alpha', \alpha), (\beta', \beta)} = \sum_i (A^i)^*_{\alpha' \beta'} A^i_{\alpha \beta} \tag{3}$$

(see Fig. 3 (a)). We denote the eigenvalues of $T[A]$ by $\lambda_i$, ordered as $|\lambda_1| \geqslant |\lambda_2| \geqslant \cdots |\lambda_{D^2}|$. If there is only one non-degenerate eigenvalue of $T[A]$ with the largest magnitude, then the tensor $A^i$ is called *injective*[3], otherwise it is called non-injective.

According to the quantum analog of Perron-Frobenius theorem, among all eigenvalues of $T[A]$ that have the largest magnitude, there must be a real one

(see Theorem 2.5 in Ref. [37] or Theorem 6.5 in Ref. [38] for more details), which we choose to be $\lambda_1$. Therefore, we can normalize $A^i$ such that $\lambda_1 = 1$, for both injective and non-injective uMPS. We will refer to such tensors as being *normalized* for convenience. If $A^i$ is normalized and injective, the norm of the state is $\langle \psi[A] | \psi[A] \rangle = \text{Tr}\left(T[A]^L\right) = 1$ in the large $L$ limit.

The eigenvalues $\{\lambda_i\}$ and the corresponding left/right eigenvectors $\{v_{i,l/r}\}$ of $T[A]$ encode important information about the entanglement and correlation of the uMPS. As we will see, when $L - N \gg 1$ and $N \gg 1$, the dominant contribution to the entanglement is related to the eigenvectors with the largest eigenvalues in magnitude. For the injective case such an eigenvector is unique, which corresponds to the eigenvalue $\lambda_1 = 1$ assuming that the tensor $A^i$ is normalized as discussed. More generally, for uMPS that may be non-injective, there can be multiple eigenvalues of modulus 1, which form the so-called peripherical spectrum. Let us denote the $i$-th left/right eigenvector corresponding to $|\lambda_i| = 1$ by $v_{i,l/r}$, i.e.,

$$|\lambda_i| = 1: \quad v_{i,l}^\mathbf{t} T[A] = \lambda_i v_{i,l}^\mathbf{t}; \ T[A] v_{i,r} = \lambda_i v_{i,r}, \quad (4)$$

These eigenvectors can always be chosen to satisfy the orthonormal condition

$$v_{i,l}^\mathbf{t} v_{j,r} = \delta_{i,j}, \tag{5}$$

because eigenvalues of modulus 1 always have trivial Jordan blocks (i.e., the nilponent part in the Jordan blocks vanishes, see Appendix C A). It turns out to be useful to reshape $v_{i,l/r}$ into matrices $v_i^{l,r}$ via

$$(v_i^l)_{\alpha' \alpha} = (v_{i,l})_{(\alpha', \alpha)}, \quad (v_i^r)_{\alpha \alpha'} = (v_{i,r})_{(\alpha', \alpha)}, \quad (6)$$

where in the left hand sides of these equations $\alpha$ and $\alpha'$ represent the row and column indices of the matrices $v_i^{l,r}$, and in the right hand sides the combination $(\alpha, \alpha')$ represents the index of the transfer matrix (as in Eq. (3)). It can be shown that if $\lambda_i = 1$, then the corresponding $v_i^l$ and $v_i^r$ are hermitian matrices which can always be chosen to be positive semi-definite (Theorem 2.5 in Ref. [37]). For simplicity, we call these eigenvectors with eigenvalues of modulus 1 *dominant eigenvectors*. Without ambiguity, we use $v^{l,r}$ without subscript to denote $v_1^{l,r}$ corresponding to $\lambda_1 = 1$.

The reason why reshaping eigenvectors $v_{i,l}$ and $v_{i,r}$ into matrices is helpful is the equivalence between the eigenvalue problem of $T[A]$ and the following problem. Given a normalized tensor $A^i$, there are two induced completely positive (CP) maps $\mathcal{E}_{A,l}$ and $\mathcal{E}_{A,r}$ [38],

$$\begin{aligned} \mathcal{E}_{A,l}(X) &= \sum_i (A^i)^\dagger X A^i; \\ \mathcal{E}_{A,r}(X) &= \sum_i A^i X (A^i)^\dagger, \end{aligned} \tag{7}$$

---

[2] In this context, a state explicitly breaks a symmetry if the expectation values of two symmetry-related local operators are different. If a state does not explicitly break a symmetry, this symmetry is spontaneously broken if there is off-diagonal long-range order, i.e., the connected two-point correlation function of two local operators that transform nontrivially under the symmetry does not decay at long distances.

[3] Strictly speaking, this is the C2-injectivity (named after the condition C2 in Ref. [29]). In the following we will refer to C2-injectivity as "injectivity" and stress the other "C1-injectivity" when necessary. These notions are reviewed in more detail in Appendix E.

and the solutions of the following equations are equivalent to the left and right eigenvectors of $T[A]$ with eigenvalue $\lambda$,

$$\mathcal{E}_{A,l}(X_l) = \lambda X_l; \quad \mathcal{E}_{A,r}(X_r) = \lambda X_r. \tag{8}$$

In particular, if $\lambda = 1$, this is a fixed-point problem of $\mathcal{E}_{A,l}$ and $\mathcal{E}_{A,r}$. Note that for a left-canonical (right-canonical) uMPS, a left (right) dominant eigenvector corresponds to the identity matrix after the reshaping in Eq. (6).

The left and right dominant eigenvectors $v_i^{l,r}$ determine the entanglement of the uMPS. Specifically, consider a subsystem with $N$ contiguous sites (e.g., the red regions in Fig. 1). Below we will focus on the limit $N \to \infty$ and $L - N \to \infty$, and in the rest of the discussions we will take these limits without clarification. We use $\rho_{\text{t.c.}}$ and $\rho_{\text{o.c.}}$ to represent the reduced density matrices of subsystem as depicted in Fig. 1, where "t.c." ("o.c.") stands for "two-cut" ("one-cut"). Note that in contrast to $\rho_{\text{t.c.}}$ which is well-defined for any uMPS tensor, $\rho_{\text{o.c.}}$ is sensitive to the boundary conditions. As discussed below Eq. (2), we always choose the boundary conditions so that the $SO(3)$ and translation symmetries are not explicitly broken in the thermodynamic limit where $L \to \infty$.

If $A^i$ is injective, which turns out to be the case we are mainly interested in, there is only one dominant eigenvalue 1, and the spectra of $\rho_{\text{t.c.}}$ and $\rho_{\text{o.c.}}$ are determined by

$$\begin{aligned}
\text{eig}(\rho_{\text{t.c.}}) &= \text{eig}((v^l v^r)^{\otimes 2}); \\
\text{eig}(\rho_{\text{o.c.}}) &= \text{eig}(v^l v^r).
\end{aligned} \tag{9}$$

The derivation of these results and their generalizations to non-injective uMPS is presented in Appendix C.

With the eigenspectra of the reduced density matrices, we can compute the entanglement entropy (von Neumann entropy) of a density matrix $\rho$,

$$S(\rho) = -\text{Tr}(\rho \ln \rho), \tag{10}$$

and the Rényi-$\alpha$ entropy ($\alpha > 0, \alpha \neq 1$),

$$S_\alpha(\rho) = -\frac{1}{\alpha - 1} \ln \text{Tr}(\rho^\alpha). \tag{11}$$

In the limit where $\alpha \to 1$, the Rényi-$\alpha$ entropy becomes the von Neumann entropy. We will simply call $S_\alpha(\rho_{\text{t.c.}})$ the two-cut Rényi entropy and $S_\alpha(\rho_{\text{o.c.}})$ the one-cut Rényi entropy. In particular, for injective uMPS, which we are mainly interested in, Eq. (9) implies

$$S_\alpha(\rho_{\text{t.c.}}) = 2S_\alpha(\rho_{\text{o.c.}}). \tag{12}$$

This relation is no longer true if the uMPS is non-injective (see Appendix C for more discussions).

Next, we move to the correlation of uMPS. The correlation length characterizes the long-distance decaying rate of correlation functions, and in uMPS it is

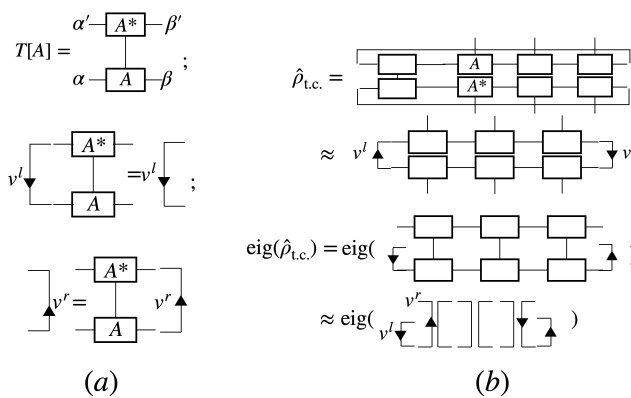

FIG. 3. (a) The schematic of transfer matrix and left/right fixed point problem. The arrows denote the direction from row index to column index. We use $\alpha, \beta$ to denote the indices of $A$ and $\alpha', \beta'$ to denote the indices of $A^*$. (b) The computation of reduced density matrix and its eigenvalues.

determined by the second largest eigenvalue (SLE) in magnitude. For simplicity, suppose there is only one SLE in the normalized $T[A]$, i.e., $1 > |\lambda_2| > |\lambda_{i>3}|$, then we insert two operators at positions 0 and $x$ in the chain of length $L$, and denote the transfer matrix with operators $\hat{O}^{(1,2)}$ inserted between the physical legs as $T^{(1,2)}$, the connected correlation function is

$$\begin{aligned}
\langle \hat{O}_0^{(1)} \hat{O}_x^{(2)} \rangle_c &= \text{Tr}\left( T[A]^{L-x-2} T^{(1)} T[A]^x T^{(2)} \right) \\
&\quad - \left( v_{1,l}^{\mathbf{t}} T^{(1)} v_{1,r} \right) \left( v_{1,l}^{\mathbf{t}} T^{(2)} v_{1,r} \right) \\
&\approx (v_{1,l}^{\mathbf{t}} T^{(1)} v_{2,r}) \lambda_2^x (v_{2,l}^{\mathbf{t}} T^{(2)} v_{1,r}).
\end{aligned} \tag{13}$$

This is an exponentially decaying function with long-distance behavior $\sim e^{-x/\xi}$, where the correlation length is $\xi = -\frac{1}{\ln |\lambda_2|}$. For more general cases where there are multiple $\lambda_i$ of modulus equal to $|\lambda_2|$, possibly with nontrivial Jordan blocks, the correlation function will receive contributions from all of them, and the scaling of $\langle \hat{O}_0^{(1)} \hat{O}_x^{(2)} \rangle_c$ is still the same for long distance $x \gg 1$.

If the uMPS tensor is non-injective, then usually there exist some local operators such that the connected correlation function is a constant at long distances. In such cases, the correlation length of the uMPS is infinite. For instance, the $p$-periodicity of states, a type of non-injectivity that will be discussed in Sec. III, indicates a spontaneously broken translation symmetry, which leads to off-diagonal long range order.

## C. Symmetric uMPS

In this subsection, we provide a synopsis of the structures of $G$-symmetric uMPS tensors with $G$ being a symmetry group, which have been discussed in literature [8, 28, 30, 39–43].

Suppose the uMPS $|\psi[A]\rangle$ is invariant under the operation $g \in G$ of a global symmetry group $G$ that has an on-site action. A unitary symmetry operation $U_g$ acting on the physical leg of the local tensor $A^i$ will lead to [30]

$$\sum_j (U_g)_{ij} A^j = e^{i\theta_g} V_g^\dagger A^i V_g \qquad (14)$$

where $V_g$ is a unitary $D \times D$ matrix. $V_g$ is in general a direct sum of some projective representations of group $G$, and the extra phase factor $e^{i\theta_g}$ is a homomorphism from from $G$ to $U(1)$. For simplicity, we assume that the group $G$ satisfies the following conditions.

(i) $G$ is non-Abelian;

(ii) $G$ is compact[4];

(iii) $G$ does not have a non-trivial one-dimensional representation.

Then the phase factor $e^{i\theta_g} = 1$ (see Appendix D). The case $G = SO(3)$ satisfies all these conditions. Other groups satisfying these conditions include simple Lie groups like $SU(n)$, $SO(n)$, etc.

The symmetry condition Eq. (14) imposes strong restrictions on the structure of $A^i$. Since the physical and bond degrees of freedom transform according to $G$, each of the physical and left/right bond Hilbert spaces can be decomposed into some irreducible representations (irreps), labelled by $\mu_p$, $\mu_a$ and $\bar{\mu}_b$, where $\bar{\mu}_b$ is the conjugate representation of $\mu_b$, as required by the transformation Eq. (14). In each representation $\mu$ of dimension $d_\mu$, we use letter $m$ ($1 \leqslant m \leqslant d_\mu$) to label the basis vectors of $\mu$. Suppose the physical degree of freedom is in the sector $\mu_p$, we can construct a multiplet of operators in the bond Hilbert space,

$$\hat{A}^{\mu_p, m_{\mu_p}} = \sum_{\mu_a, m_a, \bar{\mu}_b, \bar{m}_b} A^{\mu_p, m_{\mu_p}}_{\mu_a m_a, \bar{\mu}_b \bar{m}_b} |\mu_a, m_a\rangle \langle \bar{\mu}_b, \bar{m}_b|,$$

$$(15)$$

then the condition Eq. (14) is equivalent to saying that $\hat{A}^{\mu_p}$ forms a symmetric tensor of irrep-$\mu_p$, i.e.,

$$\hat{V}^\dagger \hat{A}^{\mu_p, m_{\mu_p}} \hat{V} = \sum_{m'_{\mu_p}} U_{m_{\mu_p}, m'_{\mu_p}} \hat{A}^{\mu_p, m'_{\mu_p}}.$$

According to Wigner-Eckart theorem (see Appendix D, or Thm 9 of Ref. [8], and Ref. [42]) and its generalizations [44], this condition implies that

$$A^{\mu_p, m_{\mu_p}}_{\mu_a, m_a, \bar{\mu}_b, \bar{m}_b} = P(\mu_p, \mu_a, \bar{\mu}_b) Q^{\mu_p, m_{\mu_p}}_{\mu_a, m_a; \bar{\mu}_b, \bar{m}_b}, \qquad (16)$$

---

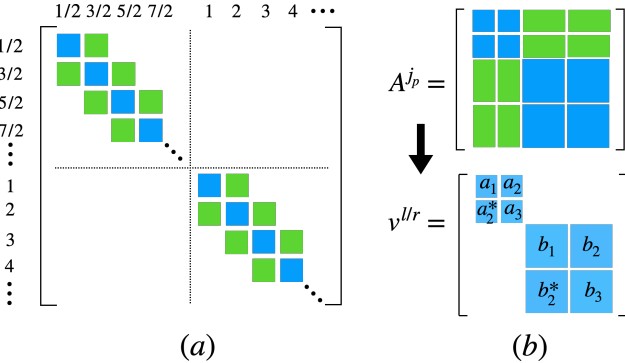

[4] The condition (ii) guarantees that $G$ can always be unitarily represented and all finite-dimensional unitary representations can be decomposed into a direct sum of irreducible representations. Groups satisfying this condition include finite groups and compact semi-simple Lie groups.

FIG. 4. (a) Structure of a symmetric tensor $A^{j_p=1}$. The blue blocks are diagonal blocks $Q_{j_a,j_a}$ and green blocks are $Q_{j_a,j_b}, |j_a - j_b| = 1$. Each small block may have some degeneracy parameters inside. (b) The symmetric tensor for $\mathbb{V}_a = j_{a,1}^{\oplus 2} \oplus j_{a,2}^{\oplus 2}$, and its eigenvector $v^{l,r}$. Every number $a_i$ ($b_i$) represents a matrix $a_i \mathbb{1}$ ($b_i \mathbb{1}$).

where $Q^{j_p, m_{\mu_p}}_{\mu_a, m_a; \bar{\mu}_b, \bar{m}_b} \equiv \langle \mu_p, m_{\mu_p}; \mu_a, m_a | \bar{\mu}_b, \bar{m}_b \rangle$ are the Clebsch-Gordan (CG) coefficients and $P(\mu_p, \mu_a, \bar{\mu}_b)$ is a constant only depending on $\mu_{p,a,b}$ (not on $m_{\mu_{p,a,b}}$).

For more general cases, the three legs may contain multiple irreducible representations, say,

$$\mathbb{V}_x = \bigoplus_{\mu_x} \left( \mathbb{V}_{\mu_x}^{\oplus d_{\mu_x}} \right), \quad x \in \{p, a, b\}, \qquad (17)$$

where each $d_{\mu_{p,a,b}}$ represents the multiplicity (or degeneracy as called in Refs. [42, 43]) of the sector $\mu_{p,a,b}$ as mentioned. Notice that the two bond subspaces $\mathbb{V}_a$ and $\mathbb{V}_b$ must be compatible, i.e., for each $\mu_a \in \mathbb{V}_a$ there is $\bar{\mu}_a$ in $\mathbb{V}_b$ and vice versa, for ensuring that the symmetry operation $V_g$ and $V_g^\dagger$ in the bond can be canceled in tensor contraction. Denote the indices of the tensor legs as $i_p = (d_{\mu_p}, \mu_p, m_{\mu_p}), i_a = (d_{\mu_a}, \mu_a, m_{\mu_a}), i_b = (d_{\mu_b}, \bar{\mu}_b, \bar{m}_{\mu_b})$, where $d_{\mu_p, \mu_a, \mu_b} \leqslant D_{\mu_p, \mu_a, \mu_b}$ labels the degeneracy index of the irrep-$\mu_{p,a,b}$. The total bond dimension is thus $D = \sum_{x=p,a,b} D_{\mu_x} d_{\mu_x}$. Then the most general expression of a symmetric tensor is

$$A^{i_p}_{i_a, i_b} = P^{\mu_p}_{\mu_a, \bar{\mu}_b} Q^{\mu_p, m_{\mu_p}}_{\mu_a, m_{\mu_a}; \bar{\mu}_b, \bar{m}_b}, \qquad (18)$$

where $P^{\mu_p}_{\mu_a, \bar{\mu}_b}$ is a rank-three tensor representing residual degrees of freedom in tensor components which are not restricted by the symmetry.

In this paper we are mainly interested in the spin rotation group $G = SO(3)$. The irreps of $SO(3)$ are labelled by integer or half-integer spin $j \in \mathbb{N}/2$ and the conjugate representation $\bar{j}$ is equivalent to $j$. In terms of matrices, $A^{j_p, m_{j_p}}$ consists of some diagonal blocks with each spin sectors $j_a = j_b$ and some off-diagonal blocks between different spin sectors $j_a \neq j_b$. For the case $j_p \in \mathbb{Z}^+$, say, $j_p = 1$, the structure of $A^{1, m_p}$ is in

the form of

$$
A^{1,m_p} = \begin{bmatrix}
B^{1,m_p}_{j_{a_1},j_{b_1}} & B^{1,m_p}_{j_{a_1},j_{b_2}} & \cdots \\
B^{1,m_p}_{j_{a_2},j_{b_1}} & B^{1,m_p}_{j_{a_2},j_{b_2}} & \cdots \\
\vdots & \vdots & \ddots
\end{bmatrix},
$$

$$
(B^{1,m_p}_{j_a,j_b})_{m_a,m_b} = P_{j_a,j_b} \otimes Q^{1,m_p}_{j_a m_a, j_b m_b},
$$

where $P_{j_a,j_b}$ is a $D_{j_a} \times D_{j_b}$ matrix of free parameters which we call the degeneracy parameters, and $Q^{1,m_p}_{j_a m_a, j_b m_b}$ is the aforementioned matrix of Clebsch-Gordan coefficients. The nonzero $(j_a, j_b)$-combinations in this case are restricted to $(j_a, j_a)$ sectors and $(j_a, j_a \pm 1)$ sectors, so the integer spin sectors and half-integer sectors are decoupled into two submatrices, and each submatrix is block tridiagonal in different spin sectors, see Fig. 4(a). For higher spin-$j_p$, the matrix will allow more off-diagonal blocks in different spin sectors, i.e., $(j_a, j_b)$ with $j_b = |j_a - j_p|, \cdots, j_a + j_p$. If $j_p$ is half-integer, the allowed blocks must satisfy $j_a \in \mathbb{Z}^+, j_b \in \mathbb{Z} + \frac{1}{2}$ or vice versa, which forces the uMPS to be non-injective [8]. We will focus on the integer $j_p$ from now on.

## III. Symmetry-enforced minimal entanglement

Equipped with the knowledge of MPS, in this section, by considering $SO(3)$-symmetric uMPS, we discuss the symmetry-enforced minimal values of $S_\alpha(\rho_{\text{t.c.}})$ and $S_\alpha(\rho_{\text{o.c.}})$ for all $\alpha \in \mathbb{R}^+$, assuming that the state in the quantum spin-$J$ chain ($J \in \mathbb{Z}$) does not explicitly or spontaneously break the $SO(3)$ or translation symmetry. We first consider the special case of von Neumann entropies, which can be viewed as the Rényi-$\alpha$ entropies with $\alpha \to 1$. We use Theorem 1 to reduce this problem to the problem of finding the symmetry-enforced minimal von Neumann entropies for *injective* $SO(3)$-symmetric uMPS, the results of which are presented in Theorem 2. The next few subsections give the proof of Theorem 2. At the end, we generalize this discussion to Rényi-$\alpha$ entropies with all $\alpha \in \mathbb{R}^+$ in Sec. III E. The analogs of these results to non-injective $SO(3)$-symmetric translation invariant uMPS, which spontaneously break the translation symmetry, are presented in Appendix H.

Below we start by presenting Theorems 1 and 2, and then we present their proofs.

**Theorem 1.** *Denote the set of all $SO(3)$-symmetric uMPS that do not spontaneously break the translation symmetry in a quantum spin-$J$ chain ($J \in \mathbb{Z}^+$) by $\mathcal{S}_J^{\text{TI}}$, and the subset of all injective $SO(3)$-symmetric uMPS by $\mathcal{S}_J^{\text{inj}} \subsetneq \mathcal{S}_J^{\text{TI}}$. Then the lower bounds of $S(\rho_{\text{t.c.}})$ and $S(\rho_{\text{o.c.}})$ in $\mathcal{S}_J^{\text{TI}}$ are the lower bounds of $S(\rho_{\text{t.c.}})$ and $S(\rho_{\text{o.c.}})$ in the subset $\mathcal{S}_J^{\text{inj}}$, respectively.*

This theorem allows us to focus on injective $SO(3)$-symmetric uMPS, whose symmetry-enforced minimal

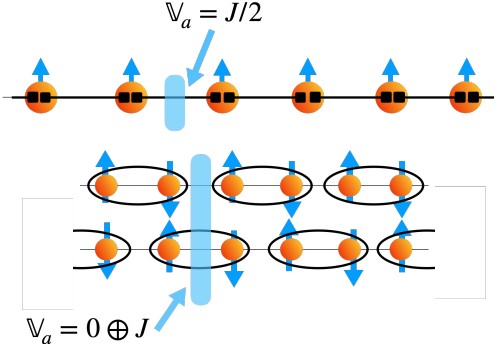

FIG. 5. $J = 1$ AKLT state (upper) and valence bond state (lower). In upper figure the pairs of black squares connected by segments are Bell pairs. In lower figure the nearest spins form singlets.

von Neumann entropies are given below.

**Theorem 2.** *In the set of injective, $SO(3)$-symmetric uMPS for a spin-$J$ chain ($J \in \mathbb{Z}^+$), the minimal entanglement entropies are lower bounded by the following relations:*

$$
\begin{aligned}
S(\rho_{\text{t.c.}}) &= 2S(\rho_{\text{o.c.}}) \geqslant S_{min}^{inj}, \\
S_{min}^{inj} &= \min\{2\ln(J+1), \ln(4(2J+1))\}.
\end{aligned}
\tag{19}
$$

*These lower bounds are tight. The uMPS with the minimal $S(\rho_{\text{o.c.}})$ is also a uMPS with the minimal $S(\rho_{\text{t.c.}})$, and vice versa.*

For $J < 7$, $2\ln(J+1) < \ln(4(2J+1))$, and for $J \geqslant 7$, $2\ln(J+1) > \ln(4(2J+1))$. The simplest injective uMPS saturating the lower bounds takes one of the following two forms

$$
\text{type-I:} \quad (A^J)^m = B^{J,m}_{\frac{J}{2},\frac{J}{2}};
$$

$$
\text{type-II:} \quad (A^J)^m = \begin{bmatrix} 0 & B^{J,m}_{0,J} \\ B^{J,m}_{J,0} & \varepsilon B^{J,m}_{J,J} \end{bmatrix}, \quad 0 < \varepsilon \ll 1
\tag{20}
$$

where $B^{J,m}_{j_1,j_2}$ is the matrix form of the CG coefficients $Q^{J,m}_{j_1,m_1;j_2,m_2}$, i.e., $\left(B^{J,m}_{j_1,j_2}\right)_{m_1 m_2} = Q^{J,m}_{j_1,m_1;j_2,m_2}$. The entanglement entropies of type-I and II states are

$$
\begin{aligned}
S\left(\rho_{\text{t.c.}}(\text{I})\right) &= 2S\left(\rho_{\text{o.c.}}(\text{I})\right) = 2\ln(J+1), \\
S\left(\rho_{\text{t.c.}}(\text{II})\right) &= 2S\left(\rho_{\text{o.c.}}(\text{II})\right) = \ln 4(2J+1) + O(\varepsilon^2).
\end{aligned}
\tag{21}
$$

For instance, the AKLT state in a spin-1 chain is of type-I, and the valence bond solid state can be approached by type-II states in the limit where $\varepsilon \to 0$ (however, the valence bond solid state itself is not a type-II state, but a non-injective MPS). A graphic presentation of their entanglement is in Fig. 5.

To prove these theorems, we define some terminologies for later use. First, an irreducible tensor is defined as follows.

**Definition 1.** *Suppose a uMPS tensor $A^i$ is normalized. $A^i$ is called irreducible if $T[A]$ has a non-degenerate eigenvalue 1 and the corresponding eigenvectors $v^l$ and $v^r$ are strictly positive definite as matrices in the bond spaces.*[5]

An irreducible tensor $A^i$ can always be transformed into the left (right) canonical form using $v^l$ ($v^r$), the matrix form of the left (right) eigenvector of the transfer matrix with eigenvalue 1. More concretely, one can verify that $\sqrt{v^r}^{-1} A^i \sqrt{v^r}$ is right-canonical, and $\sqrt{v^l} A^i \sqrt{v^l}^{-1}$ is left-canonical.

By the theorems in MPS theory (Theorems 4 and 5 in Ref. [29] or Section IV.A in Ref. [34]), given a tensor $A^i$ one can always find an equivalent standard form[6], which is a direct sum of some irreducible tensors. If the summand in this direct sum contains more than one irreducible tensors, we say that the original tensor is reducible. If some of these irreducible tensors have the same dominant weight, then the full tensor becomes non-injective. For a normalized, irreducible tensor $A_k^i$, if $T[A_k]$ has no other dominant eigenvalues other than 1 (i.e., $A_k$ is an injective uMPS), then the uMPS $A_k$ does not spontaneously break the translation symmetry; if there are other dominant eigenvalues $e^{i\theta}, \theta \in (0, 2\pi)$, then these eigenvalues will form a $\mathbb{Z}_p$ cyclic group, and the uMPS $A_k$ spontaneously breaks the translation symmetry generated by $\hat{T}$ down to its subgroup generated by $\hat{T}^p$. Such a uMPS with a spontaneously broken translation symmetry is said to have $p$-periodicity [28, 29].

Based on these facts, we can conclude that the most general form of a uMPS, where the $SO(3)$ and translation symmetries are not explicitly or spontaneously broken, is a direct sum of some injective tensors.

Now we proceed to prove Theorem 1. The proof also reduces the proof of Theorem 2 from the set of injective symmetric uMPS to a subset of irreducible injective symmetric uMPS.

*Proof of Theorem 1 and reduction of Theorem 2.*
As discussed above, any tensor $A^i \in \mathcal{S}_J^{\mathrm{TI}}$ can be put into a standard form $A^i = \oplus_{k=1}^n c_k A_k^i$, where each $A_k^i$ is normalized and irreducible. Since $A^i$ does not spontaneously break the translation symmetry, $A_k^i$ does not spontaneously break the translation symmetry. So $A_k^i$ does not have $p$-periodicity and is injective. Then by Lemma 6,

$$S(\rho_{\mathrm{t.c.}}) \geqslant \min_{k=1,2,\cdots,n} \{S(\rho_{\mathrm{t.c.}}(\rho_{A_k^i}))\}, \qquad (22)$$

where $S(\rho_{\mathrm{t.c.}}(\rho_{A_k^i}))$ is the two-cut entanglement entropy of uMPS $A_k^i$. So the lower bound of $S(\rho_{\mathrm{t.c.}})$ in $\mathcal{S}_J^{\mathrm{TI}}$ is equal to the lower bound of $\mathcal{S}_J^{\mathrm{inj}}$. The proof of $S(\rho_{\mathrm{o.c.}})$ is similar. This completes the proof of Theorem 1.

The above proof of Theorem 1 indeed reduces the proof of Theorem 2 in $S_J^{\mathrm{inj}}$ to the subset of irreducible, injective tensors in $S_J^{\mathrm{inj}}$, since each $A_k^i$ in the direct sum is irreducible.

Q.E.D.

To proceed to prove Theorem 2, we note that the form of the spin representations appearing in the bond spaces of $A^i$ also plays some role in determining the entanglement. For instance, according to the structure of a symmetric uMPS in Eq. (18), a nonvanishing spin-0 sector in the left bond space cannot appear alone. Instead, they must appear together with a spin-$J$ sector in the right bond space. In contrast, a spin-$j$ sector with $j > \frac{J}{2}$ can have diagonal blocks in $A^i$. We distinguish these two forms by the following definitions.

**Definition 2.** *Suppose that an $SO(3)$-symmetric uMPS tensor $A^i$ is irreducible. If the bond space has no spin sectors smaller than $\frac{J}{2}$, we call $A^i$ "extendable". Otherwise, we call it "generic".*

We remark that an extendable uMPS can be either injective or non-injective, and so is a generic uMPS.

The relations between different sets of symmetric uMPS described above can be presented as

$$\{\text{extendable}\} \cup \{\text{generic}\} = \{\text{irreducible}\}$$
$$\subsetneq \{\text{symmetric}\}, \qquad (23)$$
$$\{\text{extendable}\} \cap \{\text{generic}\} = \varnothing.$$

Now we can sketch the steps of prove Theorem 2:

1. Prove Proposition 1 in Sec. III A, which shows that the dominant eigenvectors of the transfer matrix associated with an irreducible symmetric uMPS can be decomposed into multiple spin-$j$ sectors. This structure will be used in the next steps.

2. Prove Theorem 2 for extendable, injective uMPS in Sec. III B.

3. Prove Theorem 2 for generic, injective uMPS in Sec. III C.

Below we carry out these steps, with some details presented in the appendices.

### A. Step 1: Structure of dominant eigenvectors

Now that we can focus on irreducible uMPS, in this subsection, we present a key proposition about the

---

[5] The name "irreducible" comes from the irreducible positive maps in $C^*$-algebra. See, for example, Ref. [37].

[6] The standard form is termed "canonical form" in literature, for instance, [29, 34]. See Appendix A for details.

dominant eigenvectors of the transfer matrix associated with these uMPS. This proposition holds not only for $SO(3)$-symmetric uMPS, but also for uMPS with more general symmetry groups discussed in Sec. II C. It states that the left and right dominant eigenvectors of the transfer matrix associated with a symmetric tensor $A^i$ have block diagonal structures in different irrep sectors. A graphic description of the block-diagonal structure with $G = SO(3)$ is shown in Fig. 4(b), for degenerate bond space $\mathbb{V}_a = j_{a,1}^{\oplus 2} \oplus j_{a,2}^{\oplus 2}$.

**Proposition 1.** *Suppose $G$ is an arbitrary symmetry group with no nontrivial one-dimensional representation (e.g., $G = SO(3)$). For a uMPS tensor $A^i$ which is irreducible and $G$-symmetric, the dominant left and right eigenvectors of its transfer matrix, after reshaped into matrices as in Eq.* (6)*, must be in the form of*

$$\bigoplus_{\mu_a \in \mathbb{V}_a} M_{\mu_a}^{l,r} \otimes \mathbb{1}_{\mu_a} \tag{24}$$

*where $\mu_a$ labels a projective representation of $G$ with dimension $d_{\mu_a}$, $\mathbb{1}_{\mu_a}$ is a $d_{\mu_a}$-dimensional identity matrix, and $M_{\mu_a}$ is a $m_{\mu_a} \times m_{\mu_a}$ matrix with $m_{\mu_a}$ the multiplicity of $\mu_a$ in $\mathbb{V}_a$.*

*Proof.* Here we prove this proposition for left dominant eigenvectors, and a similar proof can be applied to right dominant eigenvectors.

Given that $A^i$ is normalized and irreducible, the peripherical spectrum of the transfer matrix $T[A]$ is a $\mathbb{Z}_p$ finite group, each eigenvalue being non-degenerate. We need to show that all eigenvectors of each peripherical eigenvalue $e^{i\theta}$ satisfy the structure in Eq. (24). Consider any group element $g \in G$ and the corresponding unitary transformations $U_g$ and $V_g$, as well as the symmetry property of $A^i$:

$$\sum_j (U_g)_{ij} A^j = V_g^\dagger A^i V_g$$

we have the following property,

$$\begin{aligned} V_g^\dagger \mathcal{E}_{A,l}(X) V_g &= \sum_i V_g^\dagger (A^i)^\dagger V_g V_g^\dagger X V_g V_g^\dagger A^i V_g \\ &= \sum_{i,k,l} (A^k)^\dagger (V_g^\dagger X V_g) A^l (U_g^*)_{ik} (U_g)_{il} \\ &= \mathcal{E}_{A,l}(V_g^\dagger X V_g). \end{aligned} \tag{25}$$

This implies that if $X$ is an eigenvector of $\mathcal{E}_{A,l}$ with eigenvalue $\lambda$, then $V_g^\dagger X V_g$ is also an eigenvector corresponding to $\lambda$. Choosing $X$ to be the unique eigenvector corresponding to a peripherical eigenvalue $|\lambda| = 1$. Then the following relation will hold,

$$V_g^\dagger X V_g = \alpha_g X, \alpha_g \in \mathbb{C}. \tag{26}$$

It is clear that $\alpha_g$ should form a one-dimensional representation of $G$, so $\alpha_g = 1$ as we assume.

From the discussion of the structure of $A^i$, its bond space can be decomposed into a direct sum of some irreps, so the transformation $V_g$ is also a direct sum of blocks in each irrep sector. Therefore, according to the Schur's lemma.[7], as a matrix consisting of block matrices labelled by irrep sectors $(\mu_a, \bar{\mu}_b)$, $X_{\mu_a,\bar{\mu}_b}$ is nonzero only if $\mu_a = \mu_b$ and it must be proportional to identity matrix, namely $X_{\mu_a,\bar{\mu}_b} = \delta_{\mu_a,\mu_b} p_{\mu_a,\lambda} \mathbb{1}_{\mu_a}$, $p_{\mu_a,\lambda}$ being a constant. Collecting all blocks $X_{\mu_a,\bar{\mu}_b}$, we see it is in the form of Eq. (24).

Q.E.D.

We remark that although the proof of proposition 1 has used irreduciblity of tensors (which may be either injective or non-injective), this structure of dominant eigenvectors in fact holds for general reducible tensors. For instance, given a tensor which is a direct sum of two irreducible components with the same weights, its dominant eigenvectors are also in the form Eq. (24). For the detailed discussions of such cases, see Appendix G.

### B. Step 2: Extendable uMPS

In this subsection, we restrict to the subset of extendable injective uMPS in $\mathcal{S}_J^{\mathrm{inj}}$. The key feature of an extendable uMPS tensor is that it can have blocks in the diagonal spin sectors $(j, j)$ with only $j \geqslant \frac{J}{2}$, since all spin sectors in its bond space are no smaller than $\frac{J}{2}$. For such a subset, the proposition of minimal entanglement is as follows.

**Proposition 2.** *Given an irrep $j_m \geqslant \frac{J}{2}$, consider the set of extendable injective uMPS, whose bond space includes the irrep $j_m$ and all other irreps in the bond space have larger dimension than the irrep $j_m$. The minimal value of entanglement entropies that can be achieved in this set are $\ln(2j_m + 1)$ for $S(\rho_{o.c.})$ and $2\ln(2j_m + 1)$ for $S(\rho_{t.c.})$.*

*Proof.* Suppose the bond space of a uMPS tensor is $\bigoplus_i \mathbb{V}_{j_i}^{\oplus n_i}$ with $n_i$ the multiplicity of spin-$j_i$ sector. From the Proposition 1 we see that the left and right dominant eigenvector $v^{l,r}$ are block-diagonal in different spin sectors. Then, according to Eq. (9), the spectrum of reduced density matrix is in a diagonal form which is a weighted sum of some simple-spin density matrices:

$$\begin{aligned} \mathrm{eig}(\rho_{\mathrm{o.c.}}) &= \mathrm{eig}(\bigoplus_i \bigoplus_{k=1}^{n_i} t_i^{(k)} \rho_{j_i}) \\ \mathrm{eig}(\rho_{\mathrm{t.c.}}) &= \mathrm{eig}((\rho_{\mathrm{o.c.}})^{\otimes 2}), \end{aligned} \tag{27}$$

---

[7] See, e.g., Theorem 4.29 in Ref. [45] or Section 2.2 in Ref. [46].

where $\rho_{j_i} = \frac{1}{2j_i+1}\mathbb{1}_{2j_i+1}$ is only supported in $\mathbb{V}_{j_i}$. Each non-negative number $t_i^{(k)}$ is the corresponding weight of $\rho_{j_i}$ in the $k$-th $\mathbb{V}_{j_i}$, and they sum up to 1 (more precisely, these $t_i^{(k)}$ are determined by the eigenvalues of the matrices $M_{\mu_a}^{l,r}$ in Eq. (24)). Then by the concavity of the entanglement entropy (see, for example, Chapter 11.3 in Ref. [47]),

$$
\begin{aligned}
S(\rho_{\text{o.c.}}) &\geqslant \sum_i \sum_{k=1}^{n_k} t_i^{(k)} S(\rho_{j_i}) \\
&\geqslant \sum_i \sum_{k=1}^{n_k} t_i^{(k)} \ln(2j_i+1) \\
&\geqslant \ln(2j_m+1)
\end{aligned}
\tag{28}
$$

and

$$
S(\rho_{\text{t.c.}}) = 2S(\rho_{\text{o.c.}}) \geqslant 2\ln(2j_m+1). \tag{29}
$$

The equalities hold if and only if there is only a spin-$j_m$ sector in the tensor.

Q.E.D.

Based on this theorem, we can construct the minimally entangled uMPS in the set of extendable uMPS. If the physical spin is integer $J$, the smallest $j_m$ is $\frac{J}{2}$. Then the uMPS tensor $A_{j_a,j_a}^{j_p=J}$ with a simple bond space of spin $j_a = \frac{J}{2}$ will saturate the entanglement entropy $S(\rho_{\text{o.c.}}) = \ln(J+1)$ and $S(\rho_{\text{t.c.}}) = 2\ln(J+1)$. These results are explicitly verified in Sec. III D.

### C. Step 3: Generic uMPS

In the previous subsection, the minimal spin sector in the set of extendable injective uMPS provides a natural lower bound of entanglement. For the set of generic uMPS, the previous argument does not apply, since there can be valence-bond-like blocks in the tensor, say, $(j_a, j_b) = (0, J)$ and $(j_a, j_b) = (J, 0)$. The key point is that such valence-bond-like blocks will make the different spin sectors (those with spins smaller than $\frac{J}{2}$ and the corresponding ones with larger spins) in $\rho_{\text{o.c.}}$ (and in $\rho_{\text{t.c.}}$) related in some manner. Such relation will prevent the entanglement from being too small. In this subsection, we complete the proof of Theorem 2 by proving it for the generic injective uMPS.

*Proof for generic injective uMPS.* For any $j_1, j_2 < \frac{J}{2}$, $j_1 \otimes j_2$ cannot include $S$, therefore the blocks in the tensor $A^i$ are nonvanishing only for $j_a < \frac{J}{2}, j_b > \frac{J}{2}, |j_a - j_b| < S < j_a + j_b$, where $j_a(j_b)$ are in left (right) bond space or vice versa. In such cases, by Proposition 1 and Eq. (9), the spectrum of the reduced density matrix will be the spectrum of a direct sum of sectors with $j < \frac{J}{2}$ and sectors with $j > \frac{J}{2}$. We now show

that spectra of $\rho_{\text{o.c.}}$ in these two kinds of sectors are closely related. Based on this result, we can bound $S(\rho_{\text{o.c.}}) = S(\rho_{\text{t.c.}})/2$.

Suppose the tensor $A^i$ is expressed in a blocked form

$$
A^i = \begin{bmatrix} 0 & B_1^i \\ B_2^i & C^i \end{bmatrix}, \tag{30}
$$

where $C^i$ only contains blocks where $j \geqslant J/2$, and $B_{1,2}^i$ involves blocks with $j < J/2$. In the bond space of $A^i$, we denote the total dimension of spins smaller than $J/2$ by $D_1$, and the dimension of remaining spins by $D_2$. We further suppose $A^i$ is right-canonical, which does not loss any generality because $A^i$ is irreducible. Then it satisfies the following condition:

$$
\sum_i A^i (A^i)^\dagger = \mathbb{1}, \tag{31}
$$

which is equivalent to

$$
\begin{aligned}
\sum_i B_1^i (B_1^i)^\dagger &= \mathbb{1}_{D_1} \\
\sum_i B_2^i (B_2^i)^\dagger + C^i (C^i)^\dagger &= \mathbb{1}_{D_2}.
\end{aligned}
\tag{32}
$$

Now we consider the left dominant eigenvector $X$ of the transfer matrix, which is in a block diagonal form by Proposition 1. Due to Eq. (9) and the fact that $v^r = \mathbb{1}$ for a right canonical $A^i$, the eigenvalues of $X$ will be equal to the spectrum of $\rho_{\text{o.c.}}$. Explicitly, we write

$$
X = \begin{bmatrix} X_1 & \\ & X_2 \end{bmatrix}, \quad \text{Tr}(X) = 1. \tag{33}
$$

$X$ satisfies the fixed point equation of $\mathcal{E}_{A,l}(X)$,

$$
\mathcal{E}_{A,l}(X) = \sum_i (A^i)^\dagger X A^i = X, \tag{34}
$$

which is equivalent to

$$
\begin{aligned}
X_1 &= \sum_i (B_2^i)^\dagger X_2 B_2^i, \\
X_2 &= \sum_i (B_1^i)^\dagger X_1 B_1^i + (C^i)^\dagger X_2 C^i.
\end{aligned}
\tag{35}
$$

So we can rewrite $X$ as

$$
\begin{aligned}
X &= \begin{bmatrix} X_1 & \\ & \mathcal{E}_{B_1,l}(X)B_1 \end{bmatrix} + \begin{bmatrix} 0 & \\ & \mathcal{E}_{C,l}(X_1) \end{bmatrix} \\
&= p_1 \rho^{(1)} + p_2 \rho^{(2)}
\end{aligned}
\tag{36}
$$

where $p_1 + p_2 = 1, p_{1,2} > 0$ are weights of the two constitutions. $\rho^{(1)}$ and $\rho^{(2)}$ are two density matrices defined by

$$
\begin{aligned}
\rho^{(1)} &= \frac{X_1 \oplus \mathcal{E}_{B_1,l}(X_1)}{\text{Tr}\left(X_1 \oplus \mathcal{E}_{B_1,l}(X_1)\right)} \\
\rho^{(2)} &= \frac{0 \oplus \mathcal{E}_{C,l}(X_1)}{\text{Tr}\left(0 \oplus \mathcal{E}_{C,l}(X_1)\right)}.
\end{aligned}
\tag{37}
$$

Notice that, by the right-canonicality of $B_1^i$, $\mathrm{Tr}(\sum_i (B_1^i)^\dagger X_1 B_1^i) = \mathrm{Tr}(X_1)$. So if we rescale $X_1$ to $X_1'$ such that $\mathrm{Tr}(X_1') = 1$, then $\mathrm{Tr}(\mathcal{E}_{B_1,l}(X_1')) = 1$, and the spectrum of $\rho^{(1)}$ is

$$\mathrm{eig}(\rho^{(1)}) = \frac{1}{2}\mathrm{eig}\left(X_1' \oplus \mathcal{E}_{B_1,l}(X_1')\right). \qquad (38)$$

The spectrum of $X_1'$ is spin-wise by Proposition 1:

$$\mathrm{eig}(X_1') = \mathrm{eig}\left(\bigoplus_{j < \frac{J}{2}} t_j \rho_j\right), \quad \rho_j = \frac{1}{2j+1}\mathbb{1}_{2j+1}, \quad (39)$$

with $t_j$ some non-negative weights which sum up to 1. So

$$\mathrm{eig}(\rho^{(1)}) = \frac{t_0}{2}(\rho_0 \oplus \rho_J) + \sum_{0 < j \leqslant \frac{J}{2}-1} \frac{t_j}{2}\left(\rho_j \oplus \rho_{f(j)}^{mix}\right), \tag{40}$$

where $\rho_{f(j)}^{mix} = \mathcal{E}_{B_1}(\rho_j)$ is a diagonal reduced density matrix composed of spin sectors $f(j)$ which are coupled to $j$ (i.e., the block $(j, f(j))$ is non-vanishing in $A^i$).

Note that $\rho^{(1)}$ by definition is supported in both sectors of $j < \frac{J}{2}$ and sectors of spin $j \geqslant \frac{J}{2}$, while $\rho^{(2)}$ is only supported in the sectors with $j \geqslant \frac{J}{2}$. Now we can bound the entanglement entropy. By the concavity property,

$$\begin{aligned} S(\rho_{\mathrm{o.c.}}) &\geqslant p_1 S(\rho^{(1)}) + p_2 S(\rho^{(2)}) \\ &\geqslant \min\{S(\rho^{(1)}), S(\rho^{(2)})\}. \end{aligned} \qquad (41)$$

We already know that $S(\rho^{(2)}) \geqslant \ln(J+1)$ from Sec. III B. As for $\rho^{(1)}$, the particular form Eq. (40) indicates that

$$\begin{aligned} S(\rho^{(1)}) &\geqslant t_0 S\left(\frac{1}{2}\rho_0 \oplus \rho_J\right) + \sum_{0 < j \leqslant \frac{J}{2}-1} t_j S\left(\frac{1}{2}\rho_j \oplus \rho_{f(j)}^{mix}\right) \\ &\geqslant \min\{S\left(\frac{1}{2}\rho_0 \oplus \rho_J\right), S\left(\frac{1}{2}\rho_j \oplus \rho_{f(j)}^{mix}\right)\} \end{aligned} \tag{42}$$

Further we know that

$$S\left(\frac{1}{2}\rho_0 \oplus \rho_J\right) = \ln 2\sqrt{2J+1}, \qquad (43)$$

and for $0 < j \leqslant \frac{J}{2}-1$ we have the following inequality

$$\begin{aligned} &S\left(\frac{1}{2}\rho_j \oplus \rho_{f(j)}^{mix}\right) \\ =&\frac{1}{2}\left(S(\rho_j) + S(\rho_{f(j)}^{mix})\right) + \ln 2 \\ \geqslant&\frac{1}{2}\left(\ln(2j+1) + \ln(2(J-j)+1)\right) + \ln 2 \\ >&\ln 2\sqrt{2J+1}. \end{aligned} \tag{44}$$

So we see that $S(\rho^{(1)}) \geqslant \ln 2\sqrt{2J+1}$, and the equality is only achieved in the limit of non-injective uMPS with non-zero blocks $(j_1, j_2) = (0, J)$ and $(j_1, j_2) = (J, 0)$.

Therefore, we find that for generic injective uMPS,

$$S(\rho_{\mathrm{o.c.}}) \geqslant \min\{\ln(J+1), \ln 2\sqrt{2J+1}\}. \qquad (45)$$

Finally, we turn to the two-cut entropy $S(\rho_{\mathrm{t.c.}})$. In the subset of injective uMPS, the two-cut entanglement entropy $S(\rho_{\mathrm{t.c.}}) = 2S(\rho_{\mathrm{o.c.}})$, so the state with minimal $S(\rho_{\mathrm{o.c.}})$ also has the minimal $S(\rho_{\mathrm{t.c.}})$.

In summary, for generic injective uMPS,

$$\begin{aligned} S(\rho_{\mathrm{o.c.}}) &\geqslant \min\{\ln(J+1), \ln(2\sqrt{2J+1})\} \\ S(\rho_{\mathrm{t.c.}}) &\geqslant \min\{2\ln(J+1), \ln(4(2J+1))\}. \end{aligned} \qquad (46)$$

This also completes the proof of Theorem 2.

Q.E.D.

The proof in Sec. III B and Sec. III C gives the condition for an irreducible uMPS to saturate the lower bound of entanglement entropy. For the extendable case the bond space should be $\frac{J}{2}$, and for generic uMPS the bond space should be $0 \oplus J$. In the next subsection we show that these states are indeed injective and can saturate the lower bounds in Theorem 2. We remark that there can be reducible injective uMPS also saturating the symmetry-enforced lower bound of entanglement entropy, and in their standard form there is an irreducible block saturating the lower bound, and all other irreducible blocks have a small coefficient.

## D. Constructing minimally entangled states

In this subsection, we construct some explicit injective uMPS with the symmetry-enforced minimal entanglement dictated by Theorem 2.

As discussed previously, there are two classes of states which are candidates of minimally entangled states of a spin-$J$ chain:

I. $\mathbb{V}_a = \frac{J}{2}$;

II. $\mathbb{V}_a = 0 \oplus J$.

For the type-I uMPS, the tensor is just a block made of the CG coefficients:

$$(A^{J,m})_{m_a, m_b} = Q^{J,m}_{\frac{J}{2},m_a;\frac{J}{2},m_b} \qquad (47)$$

Such tensors have the following properties (see Appendix F for a summary of properties of the CG coefficients).

**Fact 1.** *Every MPS tensor $A_{j_a,j_a}^{j_p}$ with a single spin-$j_a$ is injective and $T[A]$ has a unique strictly positive left (right) eigenvector, i.e., $v^l(v^r)$ are invertible.*

The proof can be found in Appendix A of Ref. [8].

**Fact 2.** *Every MPS tensor $A_{j_a,j_a}^{j_p}$ with a single spin-$j_a$ representation in the bond space is left-canonical and right-canonical.*

*Proof.* This is due to the orthonormality relation of Clebsch-Gordan coefficients :

$$\sum_{m_1,m_2} \langle J, M | j_1, m_1; j_2, m_2 \rangle \qquad (48)$$
$$\times \langle j_1, m_1; j_2, m_2 | J', M' \rangle = \delta_{JJ'} \delta_{MM'}$$

Being left-canonical can be shown as

$$\sum_{m_{j_p}} \left( (A_{j_a,j_a}^{j_p})^\dagger A_{j_a,j_a}^{j_p} \right)_{m_2 m_2'}$$
$$= \sum_{m_{j_p},m_1} \langle j_p, m_{j_p}; j_a, m_1 | j_a, m_2 \rangle^* \langle j_p, m_{j_p}; j_a, m_1 | j_a, m_2' \rangle$$
$$= \delta_{m_2 m_2'}. \qquad (49)$$

The right-canonical can be shown as

$$\sum_{m_{j_p}} \left( A_{j_a,j_a}^{j_p} (A_{j_a,j_a}^{j_p})^\dagger \right)_{m_1 m_1'}$$
$$= \sum_{m_{j_p},m_2} \langle j_p, m_{j_p}; j_a, m_1 | j_a, m_2 \rangle \langle j_p, m_{j_p}; j_a, m_1' | j_a, m_2 \rangle^*$$
$$= \sum_{m_{j_p},m_2} \langle j_p, m_{j_p}; j_a, -m_2 | j_a, -m_1 \rangle \langle j_p, m_{j_p}; j_a, -m_2 | j_a, -m_1' \rangle$$
$$= \delta_{m_1 m_1'} \qquad (50)$$

where we used the symmetry property of CG coefficients (Eq. (126) in Appendix F)

Q.E.D.

The next claim is about the entanglement entropy carried by such a bond space with a single spin-$j$ sector.

**Fact 3.** *The uMPS with single spin-$j$ representation in the bond space has all its (half chain) Schmidt values being $\frac{1}{\sqrt{D}}$, or equivalently, all eigenvalues of reduced density matrices are $\frac{1}{D}$, where $D = 2j+1$.*

*Proof.* This follows from the canonical property in Fact 2. Since $A_{j_a,j_a}^{j_p}$ is left and right-canonical, the eigenvectors are $v^l = v^r = \mathbb{1}_{2j_a+1}$, with the normalization of state $\text{Tr}(v^l v^r) = 2j_a + 1$. So according to Eq. (9), the spectrum of reduced density matrix of a half chain is equal to the spectrum of $\frac{1}{2j_a+1} \mathbb{1}_{2j_a+1}$.

Q.E.D.

Therefore, for a type-I uMPS, $S(\rho_{\text{t.c.}}) = 2S(\rho_{\text{o.c.}}) = 2\ln(J+1)$.

The type-II uMPS takes the following form,

$$A^{J,m}(\varepsilon) = \begin{bmatrix} 0 & B_{0,J}^{J,m} \\ B_{J,0}^{J,m} & \varepsilon B_{J,J}^{J,m} \end{bmatrix} \qquad (51)$$

where $B_{j_1,j_2}^{J,m}$ is the matrix form of $Q_{j_1 m_1; j_2 m_2}^{J,m}$ with column and row indices $m_1, m_2$. $\varepsilon > 0$ is a perturbation to the tensor such that $A^{J,m}$ is injective (thus the translation symmetry is not spontaneously broken).

The following arguments show the injectivity of $A^{J,m}(\varepsilon)$ with $\varepsilon > 0$. If $\varepsilon = 0$, then the state is a valence bond state with 2-periodicity. This is because in order to have a non-vanishing normalization the chain length must be an even integer, and furthermore, the coarse-grained tensor of two sites is block diagonal, with the bond space of each block only containing one irreducible spin representation. The dominant eigenvalues of $T[A^J(\varepsilon = 0)]$ will be $\{\pm 1\}$ and both eigenvalues are non-degenerate. Under a small perturbation, the eigenvalues vary continuously with $\varepsilon$. To show that $A^{J,m}(\varepsilon)$ is injective when $\varepsilon > 0$, we just need to show that the absolute values of the eigenvalues of the transfer matrix are different when $\varepsilon > 0$.

By Proposition 1, we assume that a right eigenvector is in the matrix form

$$X_a = \begin{bmatrix} a & \\ & \mathbb{1}_{2J+1} \end{bmatrix}, \qquad (52)$$

and put it in the equation

$$\mathcal{E}_{A,r}(X_a) = \lambda_a X_a. \qquad (53)$$

With the orthogonality of Clebsch-Gordan coefficients, we see the left hand side of Eq. (53) is

$$\begin{bmatrix} 2J+1 & \\ & (\frac{a}{2J+1} + \varepsilon^2)\mathbb{1}_{2J+1} \end{bmatrix}. \qquad (54)$$

Therefore, we find an equation for $a$ and $\lambda_a$,

$$\left(\frac{a}{2J+1}\right)^2 + \varepsilon^2 \frac{a}{2J+1} - 1 = 0, \quad \lambda_a = \frac{2J+1}{a}, \quad (55)$$

which leads to

$$a_\pm = \frac{2J+1}{2}(-\varepsilon^2 \pm \sqrt{\varepsilon^4 + 4}), \quad \lambda_\pm = \frac{\pm\sqrt{\varepsilon^4+4}+\varepsilon^2}{2}. \qquad (56)$$

We see that when $0 < \varepsilon \ll 1$, $|\lambda_+| > 1 > |\lambda_-|$, which means the magnitudes of two largest eigenvalues of $T[A^J(\varepsilon)]$ are different, and there is only one dominant eigenvalue in $T[A^J(\varepsilon > 0)]$. Therefore, the injectivity of $A^{J,m}(\varepsilon)$ is proved.

With the right eigenvector $X_{a_+}$ and the corresponding left eigenvector,

$$\begin{bmatrix} \frac{1}{2}(-\varepsilon^2 + \sqrt{\varepsilon^4+4}) & \\ & \mathbb{1}_{2J+1} \end{bmatrix}$$

the entanglement entropy $S(\rho_{\text{o.c.}})$ can be shown to be

$$S(\rho_{\text{o.c.}}(\varepsilon)) = \ln(2\sqrt{2J+1}) + \frac{\varepsilon^2}{4}\ln(2J+1) + O(\varepsilon^4), \tag{57}$$

which can indeed saturate the lower bound of entanglement entropy in Theorem 2 when $\varepsilon \to 0$.

We remark that, for the type-II uMPS in Eq. (51), $S(\rho_{\text{o.c.}}(\varepsilon))$ is continuous at $\varepsilon = 0$. In contrast, $S(\rho_{\text{t.c.}}(\varepsilon))$ is not continuous: $\lim_{\varepsilon \to 0^+} S(\rho_{\text{t.c.}}(\varepsilon)) = \lim_{\varepsilon \to 0^+} 2S(\rho_{\text{o.c.}}(\varepsilon)) = \ln(4(2J+1))$, while $S(\rho_{\text{t.c.}}(\varepsilon = 0)) = \ln(2(2J+1))$ (see Appendix H B).

In conclusion, the type-I and type-II states are verified to be some minimal entangled states in the set of all injective uMPS. We remark that such states (in fact, any injective $SO(3)$-symmetric uMPS) can have parent Hamiltonians which are local, uniquely gapped, and $SO(3)$-symmetric and translation invariant. The explicit construction of such Hamiltonians is discussed in Ref. [29].

### E. Minimal Rényi entropy

In the previous subsections, we have identified the symmetry-enforced minimal entanglement entropy $S(\rho_{\text{t.c.}})$ and $S(\rho_{\text{o.c.}})$ in the set of injective uMPS, as summarized the Theorem 2. It turns out the same candidates of minimally entangled states for a given spin-$J$ chain, Eqs. (47) and (51), also exhibit the minimal Rényi entropy $S_\alpha(\rho)$ for $\alpha \in \mathbb{R}^+, \alpha \neq 1$. However, for given $J$ and $\alpha$, whether the state in Eq. (47) or the state in Eq. (51) has a smaller Rényi entropy depends on the precise values of $J$ and $\alpha$.

We summarize the results in the following theorem and give the details of the proof in Appendix I.

**Theorem 3.** *In the set $S_J^{\text{inj}}$ (the set of injective, symmetric uMPS of spin-$J$ ($J \in \mathbb{Z}^+$) chain), denote*

$$S_{\alpha,\min,t.c.}^{inj} = \min\{2\ln(J+1), -\frac{2}{\alpha-1}\ln\frac{1+\frac{1}{(2J+1)^{\alpha-1}}}{2^\alpha}\}, \tag{58}$$

*then the Rényi-$\alpha$ entropies are bounded by*

$$S_\alpha(\rho_{t.c.}) = 2S_\alpha(\rho_{o.c.}) \geqslant S_{\alpha,\min,t.c.}^{inj} \tag{59}$$

*The lower bound is a tight bound, and can approached by the type-I and type-II states in Eq. (20). Essentially, the above lower bound is also the lower bound in the set $S_J^{\text{TI}}$, namely,*

$$S_\alpha(\rho_{t.c.}) \geqslant S_{\alpha,\min,t.c.}^{inj}, S_\alpha(\rho_{o.c.}) \geqslant \frac{1}{2}S_{\alpha,\min,t.c.}^{inj} \tag{60}$$

For given $J$ and $\alpha$, Fig. 6 indicates whether the type-I or type-II state has a smaller Rényi entropy.

To prove Theorem 3, we follow a similar strategy of the proof of Theorem 2. The reduction of $S_J^{\text{TI}}$ to $S_J^{\text{inj}}$ is similar to the proof of Theorem 1 (also see Eq. (155)

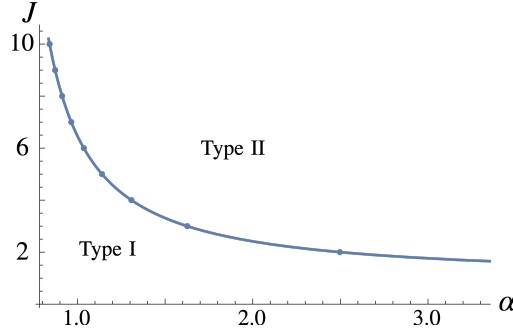

FIG. 6. Depending on the values of $J$ and $\alpha$, the minimal Rényi entropy can be saturated by either the type-I or type-II state. In this figure, for parameters below the curve the type-I state has a smaller $S_\alpha(\rho_{\text{t.c.}})$, and the for parameters above the curve the type-II state has a smaller $S_\alpha(\rho_{\text{t.c.}})$. The solid dots represent integer $J \leqslant 10$ with corresponding $\alpha$ such that both type-I and type-II states have equal $S_\alpha(\rho_{\text{t.c.}})$.

in Appendix III E). If $0 < \alpha < 1$, $S_\alpha(\rho)$ is a concave function of density matrix $\rho$ [48], so the proof can be carried out parallel to the proof of Theorem 2. For $\alpha > 1$, although $S_\alpha(\rho)$ is no longer concave, we notice that the function $H_\alpha(\rho) \equiv \text{Tr}(\rho^\alpha)$ is convex [49], so the proof of Theorem 3 can be achieved by finding the maximum of the function $H_\alpha$. Then the monotonicity of $S_\alpha$ as a function of $H_\alpha$ implies that the minimum of $S_\alpha(\rho)$ can be found at the maximum of $H_\alpha(\rho)$.

Using analysis similar to Sec. III D, it is straightforward to see that $S_\alpha(\rho_{\text{t.c.}}) = 2S_\alpha(\rho_{\text{o.c.}}) = 2\ln(J+1)$ for the type-I state, and $S_\alpha(\rho_{\text{t.c.}}) = 2S_\alpha(\rho_{\text{o.c.}}) = -\frac{2}{\alpha-1}\ln\frac{1+(2J+1)^{1-\alpha}}{2^\alpha} + \mathcal{O}(\varepsilon^2)$ for the type-II state. In Fig. 6, we show whether the type-I or type-II state has a smaller Rényi entropy as a function of $J$ and $\alpha$. In particular, we see that the AKLT state saturates the minimal value of $S_\alpha(\rho_{\text{t.c.}})$ for all $\alpha > 0$ among all spin-1 injective uMPS.

### IV. Symmetry-enforced correlation length

In this section, we turn to the correlation length of a state in a quantum spin-$J$ chain with $J \in \mathbb{Z}^+$, where neither the $SO(3)$ nor translation symmetry is explicitly or spontaneously broken. As reviewed in Sec. II B, non-injective uMPS have an infinite correlation length, and for an injective normalized uMPS, the correlation length $\xi$ is related to the second largest eigenvalue (SLE) of the transfer matrix, $\lambda_2$, via $\xi = \frac{1}{-\ln|\lambda_2|}$. It is an interesting question if there exists a symmetric injective uMPS with zero correlation length, i.e., the connected correlation function of two local operators is exactly zero as the distance between these operators becomes sufficiently large. This will happen if all eigenvalues of the transfer matrix except one of them vanish. In Ref. [34], an MPS with zero corre-

lation length is named the *renormalization group fixed point* (RGFP) of MPS. More explicitly, a tensor $A^i$ is a RGFP if $T[A] = v_{1,l}v_{1,r}^{\mathbf{t}}$.

Here we show that such symmetric RGFP does not exist for integer spin-$J$ chain (it is known to not exist if $J \in \mathbb{N}+1/2$ [8]). We present two proofs of this statement. The first proof utilizes the symmetry-enforced minimal entanglement discussed in Sec. III, and the second uses some detailed properties of the CG coefficients. After presenting the two proofs, we also provide some numerical examples of uMPS with small correlation lengths. In particular, we will see that a minimally entangled spin chain state, such as the AKLT state, does not necessarily have the minimal correlation length.

### A. Proof 1: Entanglement-based arguments

In this subsection, we show that the SLE of a symmetric uMPS cannot be zero based on the previous results of symmetry-enforced minimal entanglement.

As discussed in Ref. [34], the following are two equivalent ways to characterize the RGFP state in uMPS.

1. $T[A]^2 = T[A]$.

2. Entanglement saturation, i.e., $S(\rho_{\text{t.c.}}(N, L)) = c$ for any $N < L$.

Both of the above can be seen from the defining property that $T[A] = v_{1,r}v_{1,l}^{\mathbf{t}}$. In particular, the second notion shows that an RGFP should have $S(\rho_{\text{t.c.}}(N = 1, L \to \infty)) = \lim_{N\to\infty} \lim_{L\to\infty} S(\rho_{\text{t.c.}}(N, L))$. This is a very strong condition. To see its implication, consider a RGFP state of a spin-$J$ chain which is translation and $SO(3)$ symmetric. The reduced density matrix of a single spin must be maximally mixed, i.e., $\rho_{\text{t.c.}}(N = 1, L) = \frac{1}{2J+1}\mathbb{1}_{2J+1}$, and the corresponding two-cut von Neumann entropy is $S(\rho_{\text{t.c.}}(N = 1, L)) = \ln(2J + 1)$. But according to Theorem 2, $\lim_{N\to\infty} \lim_{L\to\infty} S(\rho_{\text{t.c.}}(N, L)) \geqslant \min\{2\ln(J + 1), \ln(4(2J + 1))\} > \ln(2J + 1)$, violating the condition $S(\rho_{\text{t.c.}}(N = 1, L \to \infty)) = \lim_{N\to\infty} \lim_{L\to\infty} S(\rho_{\text{t.c.}}(N, L))$. So we reach the following theorem.

**Theorem 4.** *Suppose a uMPS tensor $A^i$ describes an $SO(3)$-symmetric state of a quantum spin-$J$ chain $(J \in \mathbb{Z}^+)$, then the correlation length of $|\psi[A]\rangle$ cannot be exactly zero.*

### B. Proof 2: CG-coefficients-based arguments

An alternative way of showing that the correlation length cannot be exactly zero is based on the properties of the Clebsch-Gordan coefficients. As we will see, although this proof is more complicated than the previous entanglement-based argument, it provides some

estimation of lower bound of correlation length as a function of bond dimension of the uMPS.

To start, recall that from the previous discussion, if an RGFP exists, it must be an injective uMPS. So we restrict our attention to injective uMPS in this subsection. We first prove Lemma 1, a property of the Clebsch-Gordan coefficients, based on which we can also prove Theorem 4, without invoking the entanglement properties.

**Lemma 1.** $Q_{j_a,j_a}^{j_p,m_{j_p}}$ *is traceless for any positive integer* $j_p \in \mathbb{N}^+$.

*Proof.* The elements in $Q_{j_a,j_a}^{j_p,m_{j_p}}$ is defined as

$$(Q_{j_a,j_a}^{j_p,m_{j_p}})_{m_1,m_2} = \langle j_p, m_{j_p}; j_a, m_1 | j_a, m_2 \rangle \qquad (61)$$

For $m_{j_p} \neq 0$, the diagonal elements in $Q_{j_a,j_a}^{j_p,m_{j_p}}$ (i.e., the elements with $m_1 = m_2$) are always zero, so we only need to consider the $m_{j_p} = 0$ case to show the traceless-ness. The trace is then

$$
\begin{aligned}
\mathrm{Tr}(Q_{j_a,j_a}^{j_p,0}) &= \sum_m \langle j_p, 0; j_a, m | j_a, m \rangle \\
&= \sqrt{\frac{2j_a + 1}{2j_p + 1}} \sum_m (-1)^{j_a+m} \langle j_a, -m; j_a, m | j_p, 0 \rangle \\
&= \sqrt{\frac{2j_a + 1}{2j_p + 1}} \times \delta_{j_p,0}\sqrt{2j_a + 1} = 0,
\end{aligned}
$$
$$(62)$$

where the second line comes from the symmetry property

$$
\begin{aligned}
&\langle j_1, m_1; j_2, m_2 | J, M \rangle \\
&= (-1)^{j_2+m_2}\sqrt{\frac{2J + 1}{2j_1 + 1}}\langle J, -M; j_2, m_2 | j_1, -m_1 \rangle,
\end{aligned}
$$
$$(63)$$

and the last line comes from the summation property (see Ref. [50], Page 259)

$$\sum_m (-1)^{j+m}\langle j, -m; j, m | J, 0 \rangle = \delta_{J,0}\sqrt{2j + 1} \qquad (64)$$

Q.E.D.

We make two remarks here. First, Lemma 1 can be directly generalized to any half-integer $j_p \in \mathbb{N} + 1/2$, since in this case there is no $m_{j_p} = 0$. Second, the symmetry property

$$
\begin{aligned}
&\langle j_1, m_1; j_2, m_2 | JM \rangle \\
&= (-1)^{j_1+j_2-J}\langle j_1, -m_1; j_2, -m_2 | J, -M \rangle
\end{aligned}
$$
$$(65)$$

implies that, for the case of odd integer $j_p = 2k + 1$, the diagonal elements in $Q_{j_a,j_a}^{j_p,0}$ satisfies

$$\langle j_p, 0; j_a, m | j_a, m \rangle = (-1)\langle j_p, 0; j_a, -m | j_a, -m \rangle, \quad (66)$$

so the diagonal entries are anti-symmetric around the index $m = 0$. This also implies the traceless property of $Q_{j_a,j_a}^{j_p,0}$ for odd $j_p$.

Now we present a second proof of Theorem 4.

*Proof of Theorem 4.* From Eq. (18), the generic structure of $A^{j_p,m_{j_p}}$ is a block matrix, where each block is a tensor product $B^{j_p,m_{j_p}}(j_a, j_b) = P(j_a, j_b) \otimes Q_{j_a,j_b}^{j_p,m_{j_p}}$, made of the free degeneracy parameters $P(j_a, j_b)$ and the CG coefficients $Q_{j_a,j_b}^{j_p,m_{j_p}}$. From the Lemma 1 and property of tensor product $\mathrm{Tr}(A \otimes B) = \mathrm{Tr}(A)\mathrm{Tr}(B)$, the matrix $A^{j_p,m_{j_p}}$ is also traceless. The transfer matrix, defined as

$$T[A] = \sum_{m_{j_p}} (A^{j_p,m_{j_p}})^* \otimes A^{j_p,m_{j_p}} \tag{67}$$

is then also traceless. This means that the eigenvalues of $T[A]$ sum up to 0. If the uMPS is RGFP, then the trace of $T[A]$ cannot be zero, which is a contradiction.

$$\text{Q.E.D.}$$

From the proof we see if there is only one largest eigenvalue 1 in $T[A]$, then some other eigenvalues must be nonzero to cancel the largest one in trace. This immediately provides a lower bound of the second largest eigenvalue.

**Corollary 1.** *If a uMPS $A^i$ is $SO(3)$-symmetric, and the bond dimension of $A^i$ is $D$, then magnitude of the second largest eigenvalue cannot be smaller than $\frac{1}{D^2-1}$.*

*Proof.* The traceless condition implies that $\lambda_2 + \cdots + \lambda_{D^2} = -1$. On the other hand,

$$\lambda_2 + \cdots + \lambda_{D^2} \leqslant |\lambda_2| + \cdots + |\lambda_{D^2}|, \tag{68}$$

so if the magnitude of the second largest eigenvalue is smaller than $\frac{1}{D^2-1}$, the traceless condition is violated.

$$\text{Q.E.D.}$$

Notice that this lower bound $\frac{1}{D^2-1}$ may not a tight bound in general, at least the above arguments do not show it. In fact, we do not expect it to be tight, because the combination of the semi-classical picture and Lieb-Schultz-Mattis theorem in the Introduction suggests that the correlation length should diverge as $J \to \infty$, so it should be possible for the SLE to approach 1 when $J \to \infty$.

## C. Examples of states with small correlation lengths

In this subsection, we provide some numerical examples of $SO(3)$-symmetric uMPS whose SLE is locally minimal in the parameter space. The physical spin degree freedom is fixed to be $J = 1$. According

to Sec. III, the AKLT state is a minimally entangled state in this case, whose SLE is $-\frac{1}{3}$. Below we see that the presence of multiple spin sectors in the bond space can result in a decrease of the magnitude of the SLE, which in turn leads to a reduction in the correlation length. This reduction roots in the off-diagonal blocks in the transfer matrix coming from different irreps, as shown in the following examples. Indeed, both examples show an SLE with magnitude smaller than $1/3$, so both examples represent states that have smaller correlation length than the AKLT state.

*Example 1:* $\mathbb{V}_a = \frac{1}{2} \oplus \frac{3}{2}$.

In this case we have four free parameters, for $(j_a, j_b) = (\frac{1}{2}, \frac{1}{2}), (\frac{1}{2}, \frac{3}{2}), (\frac{3}{2}, \frac{1}{2}), (\frac{3}{2}, \frac{3}{2})$. We use $\boldsymbol{t} = \{t_i\}_i, 1 \leqslant i \leqslant 4$ to represent them. The local tensor $A^{1,m_p}$ is

$$A^{1,m_p} = \begin{bmatrix} t_1 B_{\frac{1}{2},\frac{1}{2}}^{1,m_p} & t_2 B_{\frac{1}{2},\frac{3}{2}}^{1,m_p} \\ t_3 B_{\frac{3}{2},\frac{1}{2}}^{1,m_p} & t_4 B_{\frac{3}{2},\frac{3}{2}}^{1,m_p} \end{bmatrix} \tag{69}$$

Since there is a overall normalization, the four free parameters $\boldsymbol{t}$ is actually in complex projective space $\mathbb{C}P^3$. Moreover, not all these parameters are physically independent, since some will be related by gauge transformations in the bond space.

Analytically minimizing the SLE here is very difficult even with this small bond dimension. So we use the `dual_annealing` method in python to numerically find a global minimum of the SLE within this parameter space, which turns out to be around `0.1061`. There are different sets of parameters $t_{1,2,3,4}$ giving rise to this SLE. For instance, one of them is

$$\begin{bmatrix} t_1 & t_2 \\ t_3 & t_4 \end{bmatrix} = \begin{bmatrix} 0.95445719 & 0.80820555 \\ 0.04342765 & 0.12444592 \end{bmatrix}.$$

*Example 2:* $\mathbb{V}_a = \left(\frac{1}{2}\right)_2 \oplus \frac{3}{2}$

In case the bond dimension is $D = 8$, and there are 9 free parameters. The local tensor is

$$A^{1,m_p} = \begin{bmatrix} t_1 B_{\frac{1}{2},\frac{1}{2}}^{1,m_p} & t_2 B_{\frac{1}{2},\frac{1}{2}}^{1,m_p} & t_5 B_{\frac{1}{2},\frac{3}{2}}^{1,m_p} \\ t_3 B_{\frac{1}{2},\frac{1}{2}}^{1,m_p} & t_4 B_{\frac{1}{2},\frac{1}{2}}^{1,m_p} & t_6 B_{\frac{1}{2},\frac{3}{2}}^{1,m_p} \\ t_7 B_{\frac{3}{2},\frac{1}{2}}^{1,m_p} & t_8 B_{\frac{1}{2},\frac{3}{2}}^{1,m_p} & t_9 B_{\frac{3}{2},\frac{3}{2}}^{1,m_p} \end{bmatrix}. \tag{70}$$

The smallest SLE is numerically found to be around `0.07624`, and an example of set of parameters that result in this SLE is

$$\begin{bmatrix} t_1 & t_2 & t_5 \\ t_3 & t_4 & t_6 \\ t_7 & t_8 & t_9 \end{bmatrix} = \begin{bmatrix} 0.33348632 & 0.70578404 & 0.66865946 \\ 0.99352508 & 0.17884033 & 0.20196437 \\ 0.30224507 & 0.15770677 & 0.16274883 \end{bmatrix}.$$

For tensors with larger bond dimensions, it is not clear whether there can an even smaller SLE than the above one. It may be interesting to search for smaller SLE in tensors with larger bond dimension and more spin sectors in the future by improving the numerical algorithms.

## V.  Discussion

In this work, we have discussed the constraints from translation and $SO(3)$ symmetries on the entanglement and correlation in integer spin chain states, based on the framework of translation invariant MPS. The main findings are summarized in Theorems 2, 3 and 4.

Although our work is based on translation invariant MPS, we expect that the symmetry-enforced minimal Rényi entropies (Theorems 2 and 3) and the impossibility of a zero correlation length (Theorem 4) actually hold for all integer spin chain states where the $SO(3)$ and translation symmetries are not explicitly or spontaneously broken. In other words, this work is based on the following two working assumptions, which we believe are reasonable but have not proved.

1. For a state in an integer spin-$J$ chain with $SO(3)$ and translation symmetries, if $S_\alpha(\rho_{\text{t.c.}}(N, L))$ and $S_\alpha(\rho_{\text{o.c.}}(N, L))$ are bounded when $(N, L - N) \to (\infty, \infty)$, then the limits $\lim_{(N,L-N)\to(\infty,\infty)} S_\alpha(\rho_{\text{t.c.}}(N, L))$ and $\lim_{(N,L-N)\to(\infty,\infty)} S_\alpha(\rho_{\text{o.c.}}(N, L))$ exist.

2. Assuming that the above limits exist, the minimal values of these limits can be achieved by a state described by a translation invariant MPS. Also, the minimal correlation length can also be achieved by such a state.

Our notion of "symmetry-enforced minimal entanglement" is based on the smallest possible values of $\lim_{(N,L-N)\to(\infty,\infty)} S_\alpha(\rho_{\text{t.c./o.c.}}(N, L))$ compatible with the symmetries, so the first assumption above is the basis for this notion to make sense. On the other hand, the second assumption is the reason why we can focus on translation invariant MPS.

Below we enumerate some interesting open questions for future studies.

1. Within the framework of uMPS, one can show that the limits $\lim_{(N,L-N)\to(\infty,\infty)} S_\alpha(\rho_{\text{t.c.}}(N, L))$ and $\lim_{(N,L-N)\to(\infty,\infty)} S_\alpha(\rho_{\text{o.c.}}(N, L))$ exist. However, as far as we know, it has not been shown in full generality that these limits exist for all translation symmetric states where $S_\alpha(\rho_{\text{t.c.}}(N, L))$ and $S_\alpha(\rho_{\text{t.c.}}(N, L))$ are bounded when $(N, L - N) \to (\infty, \infty)$. Can one prove or disprove the existence of these limits?

2. The uMPS considered in this paper can only represent states with a zero momentum. Can one prove that states with a nonzero momentum must have larger values of $\lim_{(N,L-N)\to(\infty,\infty)} S^{(\alpha)}(\rho_{\text{t.c.}}(N, L))$ and $\lim_{(N,L-N)\to(\infty,\infty)} S^{(\alpha)}(\rho_{\text{o.c.}}(N, L))$ than the ones found in Theorems 2 and 3, when these limits exist?

3. In this paper, we have shown that a state described by SO(3) symmetric uMPS cannot have a zero correlation length. But what is the precise value of the minimal correlation length? Does it indeed diverge when $J \to \infty$, as expected by combining the semiclassical analysis and Lieb-Schultz-Mattis theorem?

4. How can the considerations in this paper be generalized to multi-partite entanglement, other non-Abelian symmetry groups and/or higher dimensions?

5. Can the minimal entanglement found in this work be used for some tasks relevant to quantum information and quantum computation?

## Acknowledgments

We thank Hoi Chun Po, Yimu Bao, Zhehao Dai, Ruihua Fan, and Frank Pollmann for useful discussions. KL acknowledges the support by the National Natural Science Foundation of China/Hong Kong RGC Collaborative Research Scheme (Project No. CRS CUHK401/22). LZ is supported by the National University of Singapore start-up grants A-0009991-00-00 and A-0009991-01-00.

## Appendix A: Structure theorem of uMPS and a useful lemma

In this appendix, we review more details about the structure and reduction of uMPS, and present a useful lemma for the discussions in the main text.

As discussed in Refs. [34, 51], given any uMPS tensor, one can always get an equivalent uMPS tensor $A^i$ such that

$$A^i = \bigoplus_k c_k A_k^i \tag{71}$$

where each $A_k^i$ is irreducible and left-canonical. As described in the main text, "irreducible" means

(i) The eigenvalue 1 is non-degenerate.

(ii) After the reshaping in Eq. (6) the dominant eigenvector corresponding to eigenvalue 1 is strictly positive definite (so the reshaped matrix corresponding to this eigenvector is invertible).

(iii) There is no projection invariant subspace, i.e., there is no projector $P$ onto any proper subspace of the full bond space such that $A^i P = P A^i P$.

Conditions (i) and (ii) are defining features, and, taken together, they are equivalent to condition (iii) (for a proof, see Theorem 6.4 in Ref. [38]).

If the uMPS tensor $A^i$ is block diagonal, then the resultant wavefunction is the superposition of the states represented by each block. In this case, each block can be transformed into a normalized form independently, and the coefficients in front of each normalized block (we call them the weights of normalized forms) will determine their portion in the full quantum state. This property can be put as the following lemma.

**Lemma 2.** *Suppose a uMPS tensor has two blocks,*

$$A^i = \begin{bmatrix} c_1 A_1^i & \\ & c_2 A_2^i \end{bmatrix}, \tag{72}$$

*where $A_{1,2}$ are physically inequivalent, irreducible and normalized tensors, and $c_1 > c_2$ are two real numbers, then the largest eigenvalues of $T[A]$ and the corresponding dominant eigenvectors are determined by the largest eigenvalues and the corresponding eigenvectors of $\sum_i (A_1^i)^* \otimes A_1^i$. If $c_1 = c_2$, the largest eigenvalues and corresponding eigenvectors are determined by the largest eigenvalues and the corresponding eigenvectors of both $\sum_i (A_1^i)^* \otimes A_1^i$ and $\sum_i (A_2^i)^* \otimes A_2^i$.*

To unpack the meaning of this lemma, note that the transfer matrix $T[A]$ can be grouped into a block diagonal form of the following 4 blocks:

$$c_1^2 \sum_i (A_1^i)^* \otimes A_1^i, \ c_1 c_2 \sum_i (A_1^i)^* \otimes A_2^i,$$

$$c_2 c_1 \sum_i (A_2^i)^* \otimes A_1^i, \ c_2^2 \sum_i (A_2^i)^* \otimes A_2^i,$$

and we need to show that the largest eigenvalues will only appear in the first block if $c_1 > c_2$ (therefore the dominant eigenvector is supported only in the first block). Here we first give an argument for this lemma based on the physical requirements of state normalization, and then give a more rigorous proof.

The state $|\psi[A]\rangle$ represented by tensor $A^i$ is a superposition of two wavefunctions

$$|\psi[A]\rangle = c_1^L |\psi[A_1]\rangle + c_2^L |\psi[A_2]\rangle \tag{73}$$

with $|\psi[A_{1,2}]\rangle$ represented by $A_{1,2}^i$. The condition that $A_{1,2}^i$ are physically inequivalent indicates that $\langle \psi[A_1] | \psi[A_2] \rangle$ is exponentially suppressed as the length of chain increases, and in the thermodynamic limit the two wavefunctions are orthogonal to each other. Since $A_{1,2}^i$ are normalized, both components $|\psi[A_{1,2}]\rangle$ are also normalized. For large $L$, the relative weight $c_2^L / c_1^L$ is exponentially small, so $|\psi[A]\rangle$ is approaching $|\psi[A_1]\rangle$ in the large $L$ limit[8]. Thus in the large-$L$ limit, the normalization of the state $|\psi[A]\rangle$ is equivalent to the normalization of $|\psi[A_1]\rangle$, which exactly depends on the largest eigenvalue of $T[A_1]$. So we can conclude that the largest eigenvalues must be in the block $\sum_i (A_1^i)^* \otimes A_1^i$, and the eigenvector also follows.

Now we give a rigorous proof of Lemma 2, following the proof in Lemma. A.2 of Ref. [51].

The procedures of getting a standard form does not change the normalization of the state (and the tensor $A^i$). So without loss of generality, we can assume the CP map $\mathcal{E}_{1,2}$ of two blocks $A_1$ and $A_2$ have strictly positive definite right fixed points.

*Proof.* We now want to show that the eigenvalues of mixed transfer matrix $\sum_i (A_2^i)^* \otimes A_1^i$ are always of modulus smaller than 1, and a similar proof shows that the eigenvalues of $\sum_i (A_1^i)^* \otimes A_2^i$ are also always of modulus smaller than 1.

Denote the fixed point of $\mathcal{E}_1$ by $\Lambda_1$, which satisfies $\sum_i A_1^i \Lambda_1 (A_1^i)^\dagger = \Lambda_1$. Denote by $X$ the eigenvector of the map

$$\sum_i (A_2^i)^\dagger X A_1^i = \lambda X \tag{74}$$

with $\lambda$ an eigenvalue of $\sum_i (A_2^i)^* \otimes (A_1^i)$.

---

[8] This does not mean the state $|\psi[A]\rangle$ is equivalent to $|\psi[A]\rangle$ in the large $L$ limit. In fact, the SLE of $T[A]$ may live in other blocks in the $T[A]$, which will determine the correlation length of $|\psi[A]\rangle$.

Consider the following inequality

$$
\begin{aligned}
\left|\lambda \mathrm{Tr}(X\Lambda_1 X^\dagger)\right|^2 &= \left|\sum_i \mathrm{Tr}(XA_1^i \sqrt{\Lambda_1}\sqrt{\Lambda_1}X^\dagger (A_2^i)^\dagger)\right|^2 \\
&\leqslant \sum_i \mathrm{Tr}(XA_1^i \Lambda_1 (A_1^i)^\dagger X^\dagger) \\
&\quad \times \sum_j \mathrm{Tr}(A_2^j X\Lambda_1 X^\dagger (A_2^j)^\dagger) \\
&\leqslant \left|\mathrm{Tr}(X\Lambda_1 X^\dagger)\right|^2,
\end{aligned}
$$
(75)

where in the second line the Cauchy-Schwarz inequality is used, and in the last line we used the fact that the spectral radius of $A_2$ is 1. Since $\Lambda_1$ is positive definite, $|\mathrm{Tr}(X\Lambda_1 X^\dagger)|^2 > 0$, so $|\lambda| \leqslant 1$.

The equality $|\lambda| = 1$ holds only when $A_2^i X = \alpha XA_1^i$ for all $i$. In this case, one can further show $\alpha$ is a phase factor, since $|\lambda| = 1$ and

$$
X = \sum_i (A_2^i)^\dagger A_2^i X = \alpha \sum_i (A_2^i)^\dagger XA_1^i = \lambda\alpha X. \quad (76)
$$

Moreover, the equality $|\lambda| = 1$ requires $X$ be a unitary, meaning that $A_2$ and $A_1$ are equivalent. This can be seen as follows. Notice that

$$
\lambda^* X^\dagger X = \sum_i (A_1^i)^\dagger X^\dagger A_2^i X = \alpha \sum_i (A_1^i)^\dagger X^\dagger XA_1^i.
$$
(77)

Combining this equation with $\lambda\alpha = 1$ and $|\lambda| = 1$, we see

$$
X^\dagger X = \sum_i (A_1^i)^\dagger X^\dagger XA_1^i \quad (78)
$$

so the left-canonical property of $A_1$ requires $X^\dagger X = \mathbb{1}$ (up to a normalization), meaning $X$ is an isometry. Then we have $X^\dagger A_2^i X = \alpha A_1^i$. Denote the bond dimension of $A_1$ ($A_2$) by $D_1$ ($D_2$). Now that $X$ is an isometry, we have $D_2 \geqslant D_1$. As we assume, $A_2^i$ is irreducible, so it does not have any proper invariant subspace, which means that $D_1 = D_2$, otherwise $XX^\dagger$ will be a projector onto a proper invariant subspace of $A_2^i$. Therefore $X$ is a unitary matrix. This implies that $|\lambda| = 1$ only if $A_1^i$ is equivalent to $A_2^i$. On the other hand, if $A_1^i$ and $A_2^i$ are equivalent, then one can transform them to the same tensor without affecting the physical state, and in this case clearly $|\lambda| = 1$. Therefore, $|\lambda| = 1$ if and only if $A_1^i$ and $A_2^i$ are equivalent.

In the statement of Lemma 2, if $c_1 > c_2$, then the largest eigenvalue of $c_1 c_2 \sum_i (A_2^i)^* \otimes A_1^i$ is $c_1 c_2 \lambda$, so its absolute value is smaller than the largest eigenvalue $c_1^2$ in $T[A_1]$. If $c_1 = c_2$, since the two tensors $A_1^i$ and $A_2^i$ are inequivalent, then $\lambda < 1$ according to the above discussion, so the largest eigenvalues only lie in sectors $T[A_1]$ and $T[A_2]$.

Q.E.D.

This property is easily generalized to the case with multiple diagonal blocks.

The conclusion in Lemma 2 implies that if there are two inequivalent, irreducible blocks in $A^i$, the multiplicity of eigenvalue 1 in $T[A]$ is exactly two. This can be understood as some accidental non-injectivity, since any small perturbation in the off block-diagonal elements or the weights $c_{1,2}$ can destroy the non-injectivity.

## Appendix B: Open chain and infinite chain

In this section, we discuss the properties and boundary dependence of the open chain state defined in Eq. (2):

$$
|\psi[A], a_l, a_r\rangle = \sum_{\{i_x\}} (a_l^{i_1})^\mathbf{t} A^{i_2} \cdots A_{i_{L-1}} a_r^{i_L} |i_1 i_2 \cdots i_L\rangle
$$
(79)

We assume that $a_{l,r}^i$ are in the image of $A^i$ with $A^i$ viewed as a linear map in the bond space, so we can write $(a_l^i)^\mathbf{t} = b_l^\mathbf{t} A^i$, $a_r^i = A^i b_r$ with $b_l$ and $b_r$ two vectors. For later use we would like denote

$$
\begin{aligned}
(V[b_l])_{\alpha_0 \alpha_0'} &= (b_l)_{\alpha_0} (b_l^*)_{\alpha_0'}; \\
(V[b_r])_{\alpha_L \alpha_L'} &= (b_r)_{\alpha_L} (b_r^*)_{\alpha_L'};
\end{aligned}
$$
(80)

We would like to identify the conditions under which the $SO(3)$ and translation symmetries of the in Eq. (79) are not explicitly broken (but we allow them to be spontaneously broken).

$SO(3)$ **symmetry.** Such an MPS on a finite open chain breaks the $SO(3)$ symmetry, because symmetry transformations of the physical degrees of freedom are turned into unitaries in the bond space, and the boundary vectors are not $SO(3)$-symmetric. However, if we take the thermodynamic limit where $L \to \infty$ and consider an infinite chain, we can see the restoration of the $SO(3)$ symmetry, i.e., the expectation values of any two local operators related by an $SO(3)$ transformation are the same.

To see this, we can consider any locally supported operator $\hat{O}$, and denote the $T[\hat{O}]$ as the mixed transfer matrix. Then

$$
\langle \hat{O} \rangle = V[b_l]T[A]^{N_1}T[\hat{O}]T[A]^{N_2}V[b_r], \quad (81)
$$

where $N_{1,2}$ are some integers denoting the distances from the edges of support of $\hat{O}$ to the boundaries. For large $N_{1,2}$,

$$
\langle \hat{O} \rangle \approx \sum_{k,k'} V[b_l]v_{k,r}v_{k,l}^\mathbf{t} T[\hat{O}]v_{k',r}v_{k',l}^\mathbf{t} V[b_r] \quad (82)
$$

Applying an $SO(3)$ spin rotation to the operator $\hat{O}$, Eqs. (14) and (82) imply that the expectation value

of the transformed operator is (in the thermodynamic limit)

$$\sum_{k,k'} V[b_l](V_g^\dagger \otimes V_g) v_{k,r} v_{k,l}^{\mathbf{t}} T[\hat{O}] v_{k',r} v_{k',l}^{\mathbf{t}} (V_g \otimes V_g^\dagger) V[b_r] \tag{83}$$

where $V_g$ is the symmetry action on the bond space, such that the boundary vectors are transformed according to

$$\begin{aligned} V[b_l] &\to V[b_l](V_g^\dagger \otimes V_g) \\ V[b_r] &\to (V_g \otimes V_g^\dagger) V[b_r], \end{aligned} \tag{84}$$

So the requirement of the $SO(3)$ symmetry is that

$$\langle \hat{O} \rangle \approx \sum_{k,k'} V[b_l](V_g^\dagger \otimes V_g) v_{k,r} v_{k,l}^{\mathbf{t}} T[\hat{O}] v_{k',r} v_{k',l}^{\mathbf{t}} (V_g \otimes V_g^\dagger) V[b_r]. \tag{85}$$

Since $v_{k,l}$ and $v_{k,r}$ are dominant eigenvectors, by Proposition 1 they are singlets under the $SO(3)$ unitary $V_g^\dagger \otimes V_g$. So the expectation value $\langle \hat{O} \rangle$ is invariant under $SO(3)$. Notice this invariance is under the assumption of large $L$ and local support of operator $\hat{O}$.

Therefore, as long as the boundary vectors $a_{l,r}^i$ are in the image of $A^i$, the $SO(3)$ symmetry is not explicitly broken in the thermodynamic limit, which describes an infinite chain.

**Translation symmetry.** We also want to check if the translation symmetry is restored in an infinite chain, which can be viewed as the limit of an open chain where $L \to \infty$.

Let us consider a tensor $A^i$ of a single site in the standard form[9] in Eq. (71), and denote the inner product of the dominant eigenvectors and the boundary vectors by

$$\begin{aligned} \gamma_k &\equiv \sum_{\alpha\alpha'} V[b_l]_{\alpha\alpha'} (v_{l,r})_{\alpha'\alpha}; \\ \sigma_k &\equiv \sum_{\alpha\alpha'} (v_{k,l})_{\alpha'\alpha} V[b_r]_{\alpha\alpha'}. \end{aligned} \tag{86}$$

Eq. (82) can then be rewritten as

$$\langle \hat{O} \rangle = \sum_{k,k'} \gamma_k \sigma_{k'} (v_{k,l}^{\mathbf{t}} T[\hat{O}] v_{k',r}). \tag{87}$$

Notice because $A^i$ is in the standard form, $v_k^{l,r}$ are in a block structure (see, e.g., Eq. (98)). Moreover, any mixed transfer matrix $T[\hat{O}]$ is also block-wise: Suppose $A^i = \bigoplus A_k^i$ with each $A_k^i$ irreducible[10], then

$$(T[\hat{O}_x])_{\alpha'\alpha,\beta'\beta} = \sum_i (A^i)_{\alpha'\beta'}^* (A^j)_{\alpha\beta} (O_x)_{ij} \tag{88}$$

---

[9] In this section we do not require each block of irreducible tensor in the standard form to be left-canonical.

[10] If a tensor $A_{k,e}^i$ appears multiple times in the direct sum, we can put them together into a bigger block, which finally causes a multiplicity in the spectrum of reduced density matrix of a single copy of $A_{k,e}^i$. The relevant discussions are similar.

can be nonzero only if $\alpha$ ($\beta$) and $\alpha'$ ($\beta'$) are in the bond space of the same irreducible block. This means the $v_{k,l}$ and $v_{k',r}$ has to be in the same irreducible block such that $v_{k,l}^{\mathbf{t}} T[\hat{O}] v_{k',r}$ is nonzero.

If none of the dominant irreducible blocks has $p$-periodicity, then as long as $b_l$, $b_r$ are chosen such that for some $k$, $\gamma_k$ and $\sigma_k$ are not zero, then the translation symmetry can be restored in the large $L$ limit. This is because the expectation value of the locally supported operator does not depend on the position of the support. Notice that we have used the fact that irreducible blocks without $p$-periodicity have a unique left (right) dominant eigenvector.

If some dominant irreducible blocks in $A^i$ do have $p$-periodicity, the boundary vectors have to be chosen more suitably. Because every irreducible block lives in different bond space and they do not affect each other's dominant eigenvectors, we can simply consider the case $A^i$ has only one irreducible block with $p$-periodicity. To proceed, we first do the periodic decomposition and grouping of tensors (see Ref. [28], Ref. [38] or Appendix H). The expectation value $\langle \hat{O} \rangle$ is

$$\langle \hat{O} \rangle = \sum_{k,k'=1}^{p} \bar{\gamma}_k \bar{\sigma}_{k'} (\bar{v}_{k,l}^{\mathbf{t}} T[\hat{O}] \bar{v}_{k',r}) \tag{89}$$

where we use $\bar{v}_{k,l/r}$ to denote the eigenvectors after periodic decomposition. Now that after grouping, each $\bar{v}_{k,l/r}$ are direct sum of $p$ blocks, and $T[\hat{O}]$ is a direct sum of $p^2$ blocks. So the expectation value is indeed

$$\langle \hat{O} \rangle = \sum_{k=1}^{p} \bar{\gamma}_k \bar{\sigma}_k (\bar{v}_{k,l}^{\mathbf{t}} T[\hat{O}] \bar{v}_{k,r}) \tag{90}$$

The key point is that the translation operation $\hat{T}$ shifts the site $x$ to $x+1$, and in the periodic decomposition it transforms $T[\hat{O}_x]$ to $T[\hat{O}_{x+1}]$, so $(\bar{v}_{k,l}^{\mathbf{t}} T[\hat{O}] \bar{v}_{k,r})$ is transformed into $(\bar{v}_{k+1,l}^{\mathbf{t}} T[\hat{O}] \bar{v}_{k+1,r})$ where we identify $p+1$ with 1. However, the translation acting on the operator does not modify the boundary part, so the expectation value transforms as

$$\langle \hat{T} \hat{O} \hat{T}^{-1} \rangle = \sum_k \bar{\gamma}_k \bar{\sigma}_k (\bar{v}_{k+1,l}^{\mathbf{t}} T[\hat{O}] \bar{v}_{k+1,r}). \tag{91}$$

Now it is clear that for $\langle \hat{O} \rangle = \langle \hat{T} \hat{O} \hat{T}^{-1} \rangle$, $\bar{\gamma}_k \bar{\sigma}_k$ should be a nonzero constant independent of $k$.

In conclusion, a suitable boundary condition can be chosen as follows:

$$\text{BC: } \bar{\sigma}_k \bar{\gamma}_k = \chi > 0 \quad \text{for every } k. \tag{92}$$

Under this condition, the open chain state in Eq. (79) can restore the translation symmetry in the large $L$ limit, which can be used to describe an infinite chain.

## Appendix C: Computation of reduced density matrix

In this appendix, we derive the spectra of the reduced density matrices $\rho_{\text{t.c.}}$ and $\rho_{\text{o.c.}}$. The results are summarized in Eq. (9) in the main text, and Eqs. (96),(99) and (102) in this appendix.

### Two-cut reduced density matrices

We start with the two-cut reduced density matrices $\rho_{\text{t.c.}}$ and its spectrum.

Given a normalized tensor $A^i$, suppose there are $n$ left (right) dominant eigenvectors $v_{k,l}$ ($v_{k,r}$) corresponding to the peripherical spectrum $\{\lambda_k : |\lambda_k| = 1\}_{1 \leqslant k \leqslant n}$. As long as the length of the chain, $L$, is large enough, the normalization of the uMPS is $n$.

Now we want to trace out some physical degrees of freedom in a region (call it R) of $(L - N)$ consecutive spins (see Fig. 1), so the reduced density matrix of the complementary region is the bond-contraction of tensors in region $\bar{\text{R}}$ with their conjugation, as shown in Fig. 3(b). As shown in Appendix C A, so long as the size $L - N$ of region R is big enough, all $\bar{\text{R}}$-boundary contributions to $\hat{\rho}_{\text{t.c.}}$ is from the dominant eigenvector $v_{k,l}, v_{k,r}$ with indices rearranged (exchange the $\alpha$ and $\alpha'$ in Eq. (6)), i.e.,

$$
\begin{aligned}
\hat{\rho}_{\text{t.c.}} \propto \sum_k \sum_{\{i_1 \cdots i_N, i'_1 \cdots i'_N\}} & \left( \tilde{v}_{k,l} \rho^{i_1}_{i'_1} \cdots \rho^{i_N}_{i'_N} \tilde{v}_{k,r} \right) \\
& \times |i_1 \cdots i_N\rangle \langle i'_1 \cdots i'_N|, \\
(\tilde{v}_{k,l})_{(\alpha,\alpha')} = (v_{k,l})_{(\alpha',\alpha)}, \quad & (\tilde{v}_{k,r})_{(\alpha,\alpha')} = (v_{k,r})_{(\alpha',\alpha)},
\end{aligned}
\tag{93}
$$

where we label the $N$ spins in $\rho_{\text{t.c.}}$ by $1, \cdots N$ and denote $(\rho^{i_x}_{i'_x})_{\alpha\alpha',\beta\beta'} = A^{i_x}_{\alpha,\beta} \otimes (A^{i'_x})^*_{\alpha'\beta'}$. Notice that if $n > 1$, this expression of $\hat{\rho}_{\text{t.c.}}$ is up to a normalization, which will be determined below.

With the notation of a $d^N \times D^2$ matrix

$$
(\Psi_{\text{t.c.}})^{i_1 \cdots i_N}_{\alpha_1 \alpha_{N+1}} = (A^{i_1} A^{i_2} \cdots A^{i_N})_{\alpha_1 \alpha_{N+1}},
\tag{94}
$$

we can rewrite the matrix form of $\hat{\rho}_{\text{t.c.}}$ as

$$
\rho_{\text{t.c.}} \propto \langle i_1 \cdots i_N | \hat{\rho}_{\text{t.c.}} | i'_1 \cdots i'_N \rangle = \Psi_{\text{t.c.}} \cdot (\sum_k (v^l_k)^{\mathbf{t}} \otimes v^r_k) \cdot \Psi^{\dagger}_{\text{t.c.}}.
\tag{95}
$$

Then using a theorem in linear algebra which states that the nonzero eigenvalues of matrix product $M_1 M_2$ is equal to the nonzero eigenvalues of $M_2 M_1$, the spectrum of $\rho_{\text{t.c.}}$ can be computed as follows (see Sec. II.B.3

in Ref. [34]),

$$
\begin{aligned}
\text{eig}(\rho_{\text{t.c.}}) &\propto \text{eig}\left( \left[ \Psi^{\dagger}_{\text{t.c.}} \Psi_{\text{t.c.}} (\sum_k (v^l_k)^{\mathbf{t}} \otimes v^r_k) \right] \right) \\
&= \text{eig}\left( (T[A]^N)^{\alpha_1 \alpha_{N+1}}_{\alpha'_1 \alpha'_{N+1}} \cdot (\sum_k (v^l_k)^{\mathbf{t}} \otimes v^r_k)^{\alpha'_1 \alpha'_{N+1}}_{\beta_1 \beta_{N+1}} \right) \\
&= \text{eig}\left( \left( \sum_{k'} (v^r_{k'})^{\mathbf{t}} \otimes v^l_{k'} \right) \cdot \left( \sum_k (v^l_k)^{\mathbf{t}} \otimes v^r_k \right) \right) \\
&= \text{eig}\left( \sum_{k,k'} ((v^l_k v^r_{k'})^{\mathbf{t}} \otimes (v^l_{k'} v^r_k)) \right)
\end{aligned}
\tag{96}
$$

where "eig" means the non-zero eigenvalues. The third line is valid as long as $N$ is very large.

The exact expression of $\text{eig}(\rho_{\text{t.c.}})$, including the normalization, can be further simplified under certain circumstances. First, if $A^i$ is injective, then $n = 1$, and

$$
\text{eig}(\rho_{\text{t.c.}}) = \text{eig}\left( (v^l v^r)^{\otimes 2} \right).
\tag{97}
$$

Second, if all eigenvectors $v^{l,r}_k$ are block-diagonal and are supported in orthogonal subspaces in the bond space basis,

$$
\begin{bmatrix} v^{l,r}_1 & & \\ & 0 & \\ & & \ddots \end{bmatrix}, \begin{bmatrix} 0 & & \\ & v^{l,r}_2 & \\ & & \ddots \end{bmatrix}, \cdots, \begin{bmatrix} 0 & & \\ & \ddots & \\ & & v^{l,r}_n \end{bmatrix}
\tag{98}
$$

where we abuse the notion of $v^{l,r}_k$ and its nonzero block, then

$$
\begin{aligned}
\text{eig}(\rho_{\text{t.c.}}) &= \text{eig}\left( \frac{1}{n} \bigoplus_k (v^l_k v^r_k)^{\otimes 2} \right) \\
&= \bigoplus_k \text{eig}\left( \frac{1}{n} (v^l_k v^r_k)^{\otimes 2} \right).
\end{aligned}
\tag{99}
$$

In particular, for any non-injective tensor $A^i$, by first turning it to the standard form Eq. (71), and then doing grouping if there is any $p$-periodicity (see Ref. [28], Ref. [38] or Appendix H), we can always find the left and right eigenvectors corresponding to the final tensor in the form Eq. (98), with each $v^{l,r}_k$ being strictly positive definite. The simplified form Eq. (99) is used frequently in our proofs.

### One-cut reduced density matrices

Next, we turn to the spectrum of the one-cut reduced density matrices $\rho_{\text{o.c.}}$.

In Sec. II we mentioned that a uMPS state can be defined on an open chain as shown in Eq. (2) or Eq. (79). With suitable boundary conditions it can restore translation symmetry and $SO(3)$-symmetry in the infinite $L$ limit, which is discussed in Appendix B. We now

assume that such a suitable boundary condition is chosen, and compute the one-cut reduced density matrix by similar procedures as in Eq. (96).

Under similar to analysis as in Eq. (95), we find

$$\mathrm{eig}(\rho_{\mathrm{o.c.}}) \propto \mathrm{eig}\left(\sum_k V[b_l]v_{k,r}\Psi_{\mathrm{o.c.}} \cdot (v_k^l)^{\mathbf{t}} \cdot \Psi_{\mathrm{o.c.}}^\dagger\right) \tag{100}$$

where $(\Psi_{\mathrm{o.c.}})_\alpha^{i_1\cdots i_N} = \sum_\beta (\Psi_{\mathrm{t.c.}})_{\alpha\beta}^{i_1\cdots i_N}(b_r)_\beta \mathrm{m}$ with $b_{l,r}$ the vectors introduced below Eq. (79). Then

$$\mathrm{eig}(\rho_{\mathrm{o.c.}}) \propto \mathrm{eig}\left(\Psi_{\mathrm{o.c.}}^\dagger \Psi_{\mathrm{o.c.}}.(\sum_k \gamma_k (v_k^l)^{\mathbf{t}})\right)$$
$$\approx \mathrm{eig}\left(\sum_{k,k'}(v_{k'}^r)^{\mathbf{t}}(v_k^l)^{\mathbf{t}}\sigma_{k'}\gamma_k\right) \tag{101}$$
$$= \mathrm{eig}\left(\sum_{k,k'}(v_k^l v_{k'}^r)\sigma_{k'}\gamma_k\right),$$

where $\sigma_k$ and $\gamma_k$ are defined in Eq. (86).

For a general tensor $A^i$, this expression cannot be simplified further. If $A^i$ is in the standard form, then all irreducible tensors are in different bond spaces, so $k$ and $k'$ have to be in the same irreducible bond space in above equation to give a nonzero contribution. Therefore,

$$\mathrm{eig}(\rho_{\mathrm{o.c.}}) = \mathrm{eig}\left(\bigoplus_k \bigoplus_{\lambda_k,\lambda_k'} v_{k,\lambda_k}^l v_{k,\lambda_k'}^r \sigma_{k,\lambda_k'}\gamma_{k,\lambda_k}\right). \tag{102}$$

where $\lambda_k$ denotes the dominant eigenvalues in the $k$-th irreducible block.

The above equations can be applied to any state in the form of Eq. (79), and do not require any symmetries. For our purpose, choosing a suitable boundary condition can further simplify the above expression.

Let us consider the boundary condition in Appendix B. First, the boundary vectors $a_{l,r}^i$ are in the image of $A^i$. Second, we demand Eq. (92), which means that, after grouping tensors if there is any periodicity, the boundary vectors satisfy ($\bar{\ }$ means that the eigenvectors are corresponding to grouped tensors)

$$\mathrm{BC:}\ \bar{\sigma}_k\bar{\gamma}_k = \chi > 0 \quad \text{for every } k.$$

Under this boundary condition, both $SO(3)$ and translation symmetry will be restored in the large $L$ limit. Then the spectrum of one-cut reduced density matrix is simplified to

$$\mathrm{eig}(\rho_{\mathrm{o.c.}}) = \mathrm{eig}\left(\frac{1}{n}\bigoplus_{s=1}^n \bar{v}_s^l \bar{v}_s^r\right), \tag{103}$$

where $n$ is the total number of left dominant eigenvectors. We will ignore the symbol bar in later use.

For injective uMPS, the result is even much simpler,

$$\mathrm{eig}(\rho_{\mathrm{o.c.}}) = \mathrm{eig}(v^l v^r). \tag{104}$$

The entanglement entropy of the one-cut reduced density matrices and two-cut reduced density matrices are closely related. For injective uMPS, Eqs. (97) and (104) imply

$$S(\rho_{\mathrm{t.c.}}) = 2S(\rho_{\mathrm{o.c.}}). \tag{105}$$

For non-injective uMPS whose dominant eigenvectors fulfill the condition in Eq. (98),

$$S(\rho_{\mathrm{t.c.}}) = \frac{2}{n}S\left(\bigoplus_k (v_k^l v_k^r)\right) + \ln n;$$
$$S(\rho_{\mathrm{o.c.}}) = \frac{1}{n}S\left(\bigoplus_k (v_k^l v_k^r)\right) + \ln n, \tag{106}$$

which implies

$$S(\rho_{\mathrm{t.c.}}) = 2(S(\rho_{\mathrm{o.c.}})) - \ln n$$
$$= 2(S(\rho_{\mathrm{o.c.}}) - \ln n) + \ln n. \tag{107}$$

The last line of the above equation will be useful in the proof of minimal entanglement of non-injective uMPS, where we will search for the minimum of $S(\rho_{\mathrm{o.c.}}) - \ln n = S(\bigoplus_k(v_k^l v_k^r))$.

### A: Derivation of Eq. (93)

In this subsection, we derive Eq. (93) for completeness. We remark that Eq. (93) is valid as long as both $N$ and $L-N$ are large, and it does not require the uMPS be injective or the transfer matrix be diagonalizable.

First, we note that the reduced density matrix is given by

$$\langle i_1\cdots i_N|\hat{\rho}_{\rho_{\mathrm{t.c.}}}|i_1'\cdots i_N'\rangle$$
$$= \mathrm{Tr}_{\mathrm{R}}\left[\prod_{y\in\mathrm{R}}\rho_{j_y}^{j_y}\prod_{x\in\bar{R}}\rho_{i_x}^{i_x}\right] \tag{108}$$
$$= \sum_{\alpha\alpha'\beta\beta'}(T[A]^{L-N})_{\alpha\beta}^{\alpha'\beta'}(\prod_{x\in\bar{R}}\rho_{i_x'}^{i_x})_{\beta'\alpha'}^{\beta\alpha}$$

where $\rho_{i_x'}^{i_x} = A^{i_x}\otimes(A^{i_x'})^*$ (recall that the transfer matrix of an MPS $A^i$ is defined as $T[A] = \sum_i (A^i)^*\otimes A^i$). $\alpha,\beta$ denote the indices coming from $(A^i)^*$ and $\alpha,\beta$ denote indices coming from $A^i$ respectively. $\mathrm{Tr}_{\mathrm{R}}$ means tracing out the physical degrees of freedom in R. Notice the indices matching after taking the trace of degrees of freedom in R.

To proceed, we need to use the following structure of $T[A]$.

In general, the transfer matrix $T[A]$ can be brought into a Jordan normal form by a similarity transformation:

$$T[A] = QJQ^{-1}, J = \bigoplus_k J_k,$$

$$J_k = \begin{bmatrix} \lambda_k & 1 & & \\ & \lambda_k & 1 & \\ & & \lambda_k & \\ & & & \ddots \end{bmatrix}_{m_k \times m_k}, \tag{109}$$

where each $\lambda_k$ is an eigenvalue of $T[A]$ with multiplicity $m_k$ (there can also be multiple Jordan blocks sharing the same eigenvalue, in which case the total multiplicity of this eigenvalue is the sum of the $m_k$'s of all these blocks). As usual, we normalize the MPS so that $\lambda_1 = 1$ and order the eigenvalues such that $1 = \lambda_1 \geqslant |\lambda_2| \geqslant |\lambda_3| \geqslant \cdots$. Because $T[A]Q = QJ$, the matrix $Q$ is made of the right eigenvector and generalized eigenvectors of $T[A]$:

$$Q = (v_{1,r}^{(1)}, v_{2,r}^{(1)}, \cdots, v_{m_1,r}^{(1)}, v_{1,r}^{(2)}, v_{2,r}^{(2)}, \cdots, v_{m_2,r}^{(2)}, \cdots, v_{1,r}^{(k)}, v_{2,r}^{(k)}, \cdots, v_{m_k,r}^{(k)}, \cdots) \tag{110}$$

where

$$T[A]v_{1,r}^{(k)} = \lambda_k v_{1,r}^{(k)},$$
$$(T[A] - \lambda_k \mathbb{1})v_{i,r}^{(k)} = v_{i-1,r}^{(k)} \tag{111}$$

for $i = 2, 3, \cdots, m_k$. Similarly, because $Q^{-1}T[A] = JQ^{-1}$, the matrix $Q^{-1}$ is made of the left eigenvector and generalized eigenvectors of $T[A]^{\mathbf{t}}$:

$$Q^{-1} = (v_{m_1,l}^{(1)}, v_{m_1-1,l}^{(1)}, \cdots, v_{1,l}^{(1)}, v_{m_2,l}^{(2)}, v_{m_2-1,l}^{(2)}, \cdots, v_{1,l}^{(2)}, \cdots, v_{m_k,l}^{(k)}, v_{m_k-1,l}^{(k)}, \cdots, v_{1,l}^{(k)}, \cdots)^{\mathbf{t}}, \tag{112}$$

where

$$T[A]^{\mathbf{t}}v_{1,l}^{(k)} = \lambda_k v_{1,l}^{(k)},$$
$$(T[A]^{\mathbf{t}} - \lambda_k \mathbb{1})v_{i,l}^{(k)} = v_{i-1,l}^{(k)} \tag{113}$$

for $i = 2, 3, \cdots, m_k$.

If $m_k = 1$ for all Jordan blocks, $T[A]$ is diagonalizable and its structure greatly simplifes. Even if $T[A]$ is non-diagonalizable, the observation in Appendix B of Ref. [9] shows that $m_1 = 1$ for any physically valid MPS. Therefore, denoting the dominant eigenvectors as $v_{k,l}$ ($v_{k,r}$), we have[11]

$$T[A]^{L-N} = QJ^{L-N}Q^{-1} \approx \sum_k v_{k,r}v_{k,l}^{\mathbf{t}}. \tag{114}$$

At this point, it should be clear that Eq. (93) can be obtained reshaping the indices of Eq. (114) according to the second line of Eq. (93) and substituting the resulting equation into Eq. (108).

In passing, we note that the fact $m_1 = 1$ combined with Eqs. (110) and (112) implies that the dominant eigenvectors are orthonormalized,

$$v_{k,l}^{\mathbf{t}}v_{k',r} = \delta_{kk'}, \tag{115}$$

which is equivalent to $\mathrm{Tr}(v_k^l v_{k'}^r) = \delta_{kk'}$.

## B: Open chain and closed chain

Given a open chain state with translation symmetry in the thermodynamic limit, it may be natural to expect that deep in the bulk of the chain, the state locally looks like being in a closed chain with periodic boundary condition. More concretely, we may expect the reduced density matrix $\rho_{\mathrm{b,obc}}$ of a local region deep in the bulk of an open chain to be the same as a corresponding reduced density matrix $\rho_{\mathrm{b,pbc}}$ in the periodic chain. Below we verify this relation explicitly. Notice

---

[11] We always assume the length $L - N$ or $N$ is a multiple of $p$ when there is any $p$-periodicity, so that the factor of power of dominant eigenvalues is always 1.

the support of $\rho_{\mathrm{b}}$ is arbitrary and does not have to very large.

Now consider a tensor $A^i$ which is in the standard form and has been grouped if there is any periodicity. Use

$$(\Psi_b)_{\alpha_0\alpha_N}^{i_1\cdots i_N} = (A^{i_1}A^{i_2}\cdots A^{i_N})_{\alpha_0\alpha_N},$$

which is the same as $\Psi_{\mathrm{t.c.}}$ but we do not require $N$ to be very large. For the uMPS in the very long periodic chain, the reduced density matrix in the bulk is

$$\rho_{\mathrm{b,pbc}} = \Psi_{\mathrm{b}} \sum_k ((\bar{v}_k^l)^{\mathbf{t}} \otimes \bar{v}_k^r)\Psi_{\mathrm{b}}^\dagger, \qquad (116)$$

while for an open chain state in Eq. (79),

$$\begin{aligned}
\rho_{\mathrm{b,obc}} &= \Psi_{\mathrm{b}} \sum_{k,k'} ((v_k^l)^{\mathbf{t}} \otimes v_{k'}^r)\gamma_k\sigma_{k'}\Psi_{\mathrm{b}}^\dagger \\
&= \Psi_{\mathrm{b}} \sum_k ((v_k^l)^{\mathbf{t}} \otimes v_k^r)\gamma_k\sigma_k\Psi_{\mathrm{b}}^\dagger,
\end{aligned} \qquad (117)$$

where the summation over $k$ is for all dominant eigenvectors. The boundary condition (92) will indeed make the two reduced density matrices equal.

## Appendix D: More about symmetric tensors

In this appendix, we review more details related to Sec. II C. First we discuss why the symmetry transformation in the bond space, $V_g$, is a direct sum of projective representations, and then we present the statement of Wigner-Eckart theorem. We also present some concrete examples of symmetric tensors.

### A: Representations and Wigner-Eckart theorem

Suppose $g, h \in G$ where $G$ is the symmetry group. To see why $V_g$ can be in a projective representation, consider

$$\begin{aligned}
(U_{gh})_{ij}A^j &= (U_g)_{ik}(U_h)_{kj}A^j \\
&= (U_g)_{ik}e^{i\theta_h}V_h^\dagger A^k V_h \qquad (118) \\
&= e^{i(\theta_g+\theta_h)}V_h^\dagger V_g^\dagger A^i V_g V_h,
\end{aligned}$$

but meanwhile we know,

$$(U_{gh})_{ij}A^j = e^{i\theta_{gh}}V_{gh}^\dagger A^i V_{gh}. \qquad (119)$$

So if we have $V_g V_h = \omega(g, h)V_{gh}$ where $\omega(g, h)$ is a phase factor satisfying

$$\omega(g_1, g_2)\omega(g_1 g_2, g_3) = \omega(g_1, g_2 g_3)\omega(g_2, g_3), \qquad (120)$$

then Eqs. (118) and (119) are satisfied and consistent. In such case, $V_g$ form a projective representation.

In general, $V_g$ may be a direct sum of some irreducible projective representations (this direct sum itself may not be a projective representation), and $\omega(g, h)$ can be a direct sum of identity matrices multiplied by the phase factors related to the projective representations in $V_g$.

On the other hand, the phase $e^{i\theta_g}$ satisfies $e^{i\theta_g}e^{i\theta_h} = e^{i\theta_{gh}}$, so this is an homomorphism from $G$ to $U(1)$. For the case we are interested in, $G$ is the $SO(3)$ spin rotation group, and the only homomorphism from $SO(3)$ to $U(1)$ is the trivial one, so $e^{i\theta_g} = 1$ by the first isomorphism theorem[12]. Our discussion will be focused on $G = SO(3)$ from now on, but it is straightforwardly generalized to other symmetry groups.

Next, we present some details of Wigner-Eckart theorem. Consider an operator $\hat{T}_m^j$ in spin-$j$ representation, i.e. $\hat{U}\hat{T}_m^j\hat{U}^\dagger = \sum_{m'=-j}^j \hat{T}_{m'}^j D_{m'm}^j$ where $\hat{U}$ is an operator of $SU(2)$ transformation acting on the basis of the Hilbert space $\hat{T}_m^j$ lives in (i.e., the bond space in our case) and $D_{m'm}^j$ is the Wigner D-matrix of spin-$j$. The Wigner-Eckart theorem states that the elements of $\hat{T}_m^j$ are in the form $\langle j_1, m_1|\hat{T}_m^j|j_2, m_2\rangle = \langle j_1, m_1; j, m|j_2, m_2\rangle\langle j_1||\hat{T}_m^j||j_2\rangle$ where $\langle j_1||\hat{T}_m^j||j_2\rangle$ is independent of $m, m_1, m_2$.

The Wigner-Eckrat theorem was originally proposed for the $SO(3)$ group, and it can be generalized to other groups satisfying the complete reducibility of finite dimensional representations. For instance, analogue of the previous statement can be proved for finite groups, compact groups and semisimple Lie groups. Interested readers are referred to Ref. [44] for details.

### B: Examples

In this subsection, we present some examples of symmetric tensors.

*Example 1: $j_p = 1, j_a = j_b = 1/2$.* This choice of $j_{a,b}$ is the allowed bond space with the smallest dimension for spin-1 chain. The fusion rule involved here is $\frac{1}{2}\otimes 1 = \frac{1}{2} \oplus \frac{3}{2}$. The Clebsch-Gordan coefficients are organized into $Q$-factors

$$Q^1 = \frac{1}{\sqrt{3}}\begin{bmatrix} 0 & 0 \\ \sqrt{2} & 0 \end{bmatrix}, \quad Q^0 = \frac{1}{\sqrt{3}}\begin{bmatrix} -1 & 0 \\ 0 & 1 \end{bmatrix},$$
$$Q^{-1} = \frac{1}{\sqrt{3}}\begin{bmatrix} 0 & -\sqrt{2} \\ 0 & 0 \end{bmatrix}. \qquad (121)$$

The row and column indices of $Q$ are ordered as decreasing $m_j$ from $m_j = j$, here just $[1/2, -1/2]$. In

---

[12] More concretely, $G/\mathrm{ker}f \cong \mathrm{im}(f) \subset H$ for a homomorphism $f : G \to H$. The $\mathrm{ker}f$ is a normal group of $G$. But $SO(3)$ is simple having no nontrivial normal subgroup, thus $f$ has to be trivial.

this case there is no extra free parameter except an overall constant, so $A^{m_p} = const. \times Q^{m_p}$. In fact this is the unique uMPS of bond dimension $D = 2$ for a translation-invariant, SO(3)-symmetric spin-1 chain (up to some bond-space gauge transformations which do not change the physical correlations). In fact, this state is exactly the ground state of the renowned AKLT model [35, 36].

*Example 2:* Bond space with a single $j_a$. These tensors can be thought of as generalizations of the AKLT state from the MPS perspective. Each $A^{j_p,m_p}$ is equal to $Q^{j_p,m_p}_{j_a,j_a}$. These states also have no free parameters. For instance, $j_p = 1, j_a = 1$, the matrices are

$$Q^1 = \begin{bmatrix} 0 & 0 & 0 \\ \frac{1}{\sqrt{2}} & 0 & 0 \\ 0 & \frac{1}{\sqrt{2}} & 0 \end{bmatrix}, Q^0 = \begin{bmatrix} -\frac{1}{\sqrt{2}} & 0 & 0 \\ 0 & 0 & 0 \\ 0 & 0 & \frac{1}{\sqrt{2}} \end{bmatrix},$$
$$Q^{-1} = \begin{bmatrix} 0 & -\frac{1}{\sqrt{2}} & 0 \\ 0 & 0 & -\frac{1}{\sqrt{2}} \\ 0 & 0 & 0 \end{bmatrix}. \tag{122}$$

The entanglement of uMPS with single-$j_a$ bond is simple, which we discuss in Sec. III. The correlation length, by some observation of examples, obeys the following rules.

*Observation 1:* For the spin-1 uMPS, the second largest eigenvalue of $T[A^{j_p=1}_{j_a,j_a}]$ is of magnitude $\left|1 - \frac{1}{j_a(j_a+1)}\right|$, regardless of $j_a$ being integer or half-integer.

*Observation 2:* For the spin-$J$ uMPS with bond space $\mathbb{V}_a = \frac{J}{2}$, $J \in \mathbb{Z}$ the second largest eigenvalue of $T[A^{j_p=J}_{j_a=\frac{J}{2},j_a=\frac{J}{2}}]$ is of magnitude $1 - \frac{2}{(J+2)}$.

## Appendix E: C1- and C2-injectivity

In this appendix, we review the distinction and relation between the C1-injectivity and C2-injectivity [29]. Define a map $\Gamma_{\tilde{A}_l}$ from the matrix space $\mathcal{M}_D$ to the physical Hilbert space of dimension $d^l$,

$$\Gamma_{\tilde{A}_l}(X) = \sum_{\{i_x\}} \text{Tr}[X\tilde{A}^{i_1 \cdots i_l}] |i_1 \cdots i_l\rangle \tag{123}$$

where $X$ is a $D \times D$ matrix and $\tilde{A}^{i_1 \cdots i_l}_l = A^{i_1} \cdots A^{i_l}$. A tensor $A^i$ is C1-injective if $\Gamma_{\tilde{A}_l}$ is an injective map for every integer $l > l_0$, and $A^i$ is C2-injective if $T[A]$ has $1 = \lambda_1 > |\lambda_{i \geqslant 2}|$.

The following proposition shows that C1-injectivity is stronger than C2-injectivity.

**Proposition 3.** *Suppose there is a tensor $A^i$, the following two statements are equivalent:*

(i) $A^i$ is C1-injective;

(ii) $A^i$ is C2-injective, and its dominant eigenvector is strictly positive definite.

*Proof.* (i)$\Rightarrow$ (ii) is straightforward. Being C1-injective requires the standard form Eq. (71) has only one block, otherwise any X with only off-diagonal blocks will lie in the kernel of map $\Gamma_{\tilde{A}_l}$ for every integer $l$. Then the C2-injectivity and strictly positive dominant eigenvector follows from the properties of block in the standard form.

(ii)$\Rightarrow$(i) is more complicated and is proved in Lemmas 5.2 and 5.3 in Ref. [28] (also Theorem 6.7 (4$\rightarrow$1) in Ref. [38]).

Q.E.D.

## Appendix F: Properties of CG coefficients

We summarize some properties of the CG coefficients of the $SO(3)$ group for reference (see Chapter 8 of Ref. [50], for example). First, CG coefficients are real, i.e.,

$$\langle j_1, m_1; j_2, m_2 | J, M \rangle = \langle J, M | j_1, m_1; j_2, m_2 \rangle. \tag{124}$$

Then the orthogonality relation:

$$\sum_{m_1,m_2} \langle J, M | j_1, m_1; j_2, m_2 \rangle$$
$$\times \langle j_1, m_1; j_2, m_2 | J', M' \rangle = \delta_{JJ'}\delta_{MM'}$$
$$\sum_{J=|j_1-j_2|}^{j_1+j_2} \sum_{M=-J}^{J} \langle j_1, m_1; j_2, m_2 | J, M \rangle$$
$$\times \langle J, M | j_1, m_1; j_2, m_2 \rangle = \delta_{m_1 m_1'}\delta_{m_2 m_2'} \tag{125}$$

The symmetry properties:

$$\langle j_1, m_1; j_2, m_2 | JM \rangle$$
$$= (-1)^{j_1+j_2-J}\langle j_1, -m_1, j_2, -m_2 | J, -M \rangle$$
$$= (-1)^{j_1+j_2-J}\langle j_2, m_2; j_1, m_1 | J, M \rangle$$
$$= (-1)^{j_1-m_1}\sqrt{\frac{2J+1}{2j_2+1}}\langle j_1, m_1; J, -M | j_2, -m_2 \rangle$$
$$= (-1)^{j_2+m_2}\sqrt{\frac{2J+1}{2j_1+1}}\langle J, -M; j_2, m_2 | j_1, -m_1 \rangle \tag{126}$$

And summation property:

$$\sum_m (-1)^{j+m}\langle j, -m; j, m | J, 0 \rangle = \delta_{J,0}\sqrt{2j+1} \tag{127}$$

## Appendix G: Generalizations of Proposition 1

In section III A we have proved Proposition 1, stating that the dominant eigenvectors of irreducible tensors, after the reshaping in Eq. (6), are singlets in each spin sectors. In this appendix, we show that the singlet structure also holds for reducible tensors which is equivalent to a direct sum of some irreducible tensors.

Consider the case where the tensor $A^i$ contains a few dominant blocks, each of which is irreducible. We only need to show that each irreducible block is $G$-symmetric, then by Proposition 1, each irreducible block will have eigenvectors of peripherical spectrum in the form of Eq. (24), so Proposition 1 is generalized to reducible tensors.

Suppose we have two such irreducible blocks which are not equivalent (i.e., there is no similarity transformation connecting the two),

$$A^i = \begin{bmatrix} A_1^i & \\ & A_2^i \end{bmatrix}, \qquad (128)$$

and consider an arbitrary symmetry transformation of physical degrees of freedom $U_g$. Denote

$$\tilde{A}_{1,2}^i = \sum_j (U_g)_{ij} A_{1,2}^j,$$

then the mixed transfer matrix $T_U[A]$ defined below

$$T_U[A] = \sum_{i,j} (U_g)_{ij} (A^i)^* \otimes A^j \qquad (129)$$

will decompose into a direct sum of four pieces

$$T_U[A_1] \oplus T_U[A_1^*, A_2] \oplus T_U[A_2^*, A_1] \oplus T_U[A_2], \quad (130)$$

where we use the notation

$$T_U[A_k^*, A_{k'}] = \sum_{i,j} (U_g)_{ij} (A_k^i)^* \otimes A_{k'}^j.$$

Since $A^i$ is symmetric, $T_U[A]$ must have two sets of peripherical eigenvalues (as the action $U$ goes to the bond space and do not influence the eigenvalues), and $T_U[A_1^*, A_2]$ and $T_U[A_2^*, A_1]$ cannot have spectral radius 1, by a smilar argument as in Appendix A. The two sets of peripherical spectra must come from $T_U[A_1]$ and $T_U[A_2]$. Suppose $T_U[A_1]$ has a set of peripherical spectra, then by the Lemma 1 in Ref. [30], $A_1$ must be $G$-symmetric. This also requires $A_2$ to be $G$-symmetric, by the symmetry of $|\psi[A]\rangle$

$$\left|\psi[A^i]\right\rangle = \left|\psi[A_1^i]\right\rangle + \left|\psi[A_2^i]\right\rangle = \left|\psi[\tilde{A}_1^i]\right\rangle + \left|\psi[\tilde{A}_2^i]\right\rangle.$$

We can further consider the reducible tensor $A^i$ with more than two irreducible components $A_{(k)}^i, 1 \leqslant k \leqslant n$. We first group all blocks $A_{k>1}^i$ as a tensor $A'$, and analysis above shows that $A_1^i$ and $A'^i$ are both symmetric. In particular, the transfer matrix of $A'^i$ has spectral radius 1. Iterating the analysis for $A'$, we finally conclude that all irreducible blocks $A_{(k)}^i$ are symmetric.

## Appendix H: Entanglement of non-injective uMPS

In Sec. III C, we have discussed the proof of the minimal entanglement for generic injective uMPS. We have also seen from Eq. (20) that the type-II states can approach non-injective uMPS when $\varepsilon \to 0$, which spontaneously break the translation symmetry. In this appendix, we give more details about the entanglement of such translation-symmetry-breaking states. These results complete the understanding of the symmetry-enforced minimal entanglement for *all* $SO(3)$-symmetric uMPS, rather than only those without spontaneous symmetry breaking.

For simplicity, suppose $A^i$ is irreducible, i.e., there is only one block in the standard form Eq. (71) of the non-injective uMPS. According to the MPS theories (Proposition 3.3 in Ref. [28] or Theorem 6.6 in Ref. [38]), the peripherical spectrum is a finite $\mathbb{Z}_p$ group, consisting of all $p$-th roots of unity for some positive integer $p$. This uMPS admits a *periodic decomposition* [29], i.e., there are $p$ different tensors $A_1^i, \cdots, A_p^i$ constructed from the projection $A_k^i = P_k A^i P_{k+1}$, where $P_k$ is the projection to a subspace of the bond space, such that the original uMPS is equivalent to the uMPS of a clustered tensor $\tilde{A}^{i_1 i_2 \cdots i_p}$ defined below. More precisely, let

$$\begin{aligned} \tilde{A}_k^{i_1 i_2 \cdots i_p} &\equiv A_k^{i_1} A_{k+1}^{i_2} \cdots A_p^{i_{p-k+1}} A_1^{i_{p-k+2}} \cdots A_{k-1}^{i_p}, \\ \tilde{A}^{i_1 i_2 \cdots i_p} &\equiv \bigoplus_k \tilde{A}_k^{i_1 i_2 \cdots i_p}, \end{aligned} \qquad (131)$$

then the original MPS $|\psi[A]\rangle$ can be identified as

$$|\psi[A]\rangle = |\psi[\tilde{A}]\rangle, \qquad (132)$$

where $|\psi[\tilde{A}]\rangle$ is viewed as an MPS defined on a lattice formed by enlarged sites, with each enlarged site being $p$ consecutive sites of the original lattice, and the basis of each enlarged site is the tensor product of the basis of $p$ original sites, i.e., $|i_1 i_2 \cdots i_p\rangle$.

It is also useful to think of such $|\psi[A]\rangle = |\psi[\tilde{A}]\rangle$ as a superposition of $p$ states which are related by translation,

$$|\psi[A]\rangle = \sum_k \left|\psi[\tilde{A}_k]\right\rangle, \hat{T}\left|\psi[\tilde{A}_k]\right\rangle = \left|\psi[\tilde{A}_{k+1}]\right\rangle.$$

Each component $\left|\psi[\tilde{A}_k]\right\rangle$ is $SO(3)$-symmetric and symmetric under $p$-site translation $\hat{T}^p$, and it can also be viewed as an MPS defined on a lattice with the enlarged sites.

For our purpose, we only need the existence of such a periodic decomposition, and we do not have to explicitly construct $P_k$. Readers interested in constructing $P_k$ are referred to Ref. [29] for details.

## A: Some lemmas

Below we first present multiple useful lemmas.

**Lemma 3.** *Given a density matrix $\rho$, and a right-canonical tensor $A^i$ together with the induced CP map $\mathcal{E}_{A,l}$*

$$\mathcal{E}_{A,l}(\rho) = \sum_i (A^i)^\dagger \rho A^i,$$

*then the image $\mathcal{E}_{A,l}(\rho)$ is still a density matrix.*

*Proof.* We need to show that $\mathcal{E}_{A,l}(\rho)$ is positive and has trace 1. The positivity is clear by definition. Since $A^i$ is right-canonical, $\sum_i A^i(A^i)^\dagger = \mathbb{1}$, $\mathcal{E}_{A,l}$ is indeed a trace-preserving map, since

$$\mathrm{Tr}\left[\sum_i (A^i)^\dagger \rho A^i\right] = \mathrm{Tr}\left[\rho\right] = 1. \qquad (133)$$

So $\mathcal{E}_{A,l}(\rho)$ is a density matrix.

Q.E.D.

**Lemma 4.** *Given a normalized p-periodic tensor $A^i$ and the periodic decomposition $A^i_k$ and $\tilde{A}^{i_1\cdots i_p}_k$ as discussed in Eq. (131), suppose $X_k$ is the dominant left eigenvector of $T[\tilde{A}_k]$ for $1 \leqslant k < p$, then the dominant left eigenvector of $T[\tilde{A}_{k+1}]$ satisfies*

$$X_{k+1} = \sum_i (A^i_k)^\dagger X_k A^i_k \qquad (134)$$

*up to multiplication by a constant.*

*Proof.* Because $X_k$ is left eigenvector of $T[\tilde{A}_k]$,

$$X_k = \sum_{i_1\cdots i_p} (\tilde{A}^{i_1\cdots i_p}_k)^\dagger X_k \tilde{A}^{i_1\cdots i_p}_k. \qquad (135)$$

So

$$\sum_i (A^i_k)^\dagger X_k A^i_k$$
$$= \sum_{i_2\cdots i_p i} (\tilde{A}^{i_2\cdots i_p i}_{k+1})^\dagger \left(\sum_{i_1} (A^{i_1}_k)^\dagger X_k A^{i_1}_k\right) \tilde{A}^{i_2\cdots i_p i}_{k+1}, \qquad (136)$$

which implies $\sum_i (A^i_k)^\dagger X_k A^i_k$ is the dominant left eigenvector of $T[\tilde{A}_{k+1}]$.

Q.E.D.

Furthermore, if every $\tilde{A}^{i_1\cdots i_p}_k$ is right-canonical, we can choose the proper normalizations for each $A^i_k$ such that each $A^i_k$ satisfies the right-canonical condition

$$\sum_i A^i_k (A^i_k)^\dagger = \mathbb{1}.$$

To see this, consider

$$\sum_{i_1\cdots i_p} \tilde{A}^{i\cdots i_p}_1 (\tilde{A}^{i\cdots i_p}_1)^\dagger = \mathbb{1}$$
$$\Rightarrow \sum_{ii_1\cdots i_p} A^i_p A^{i\cdots i_p}_1 (\tilde{A}^{i\cdots i_p}_1)^\dagger (A^i_p)^\dagger = \sum_i A^i_p (A^i_p)^\dagger, \qquad (137)$$

so by the uniqueness of the left eigenvector of $T[\tilde{A}_p]$ we see $\sum_i A^i_p(A^i_p)^\dagger = \alpha_p \mathbb{1}$. Similar manipulations lead to $\sum_i A^i_k(A^i_k)^\dagger = \alpha_k \mathbb{1}$ for $1 \leqslant k \leqslant p$ and $\alpha_1\alpha_2\cdots\alpha_p = 1$. It is always possible to choose $\alpha_k = 1$ for every $k$. Based on this choice, $X_k$ satisfies $\mathrm{Tr}(X_k) = 1$ and is thus a valid density matrix, then $\sum_i (A^i_k)^\dagger X_k A^i_k$ is also a valid density matrix.

The next lemma is useful in the discussion of the entanglement in $p$-periodic states. Suppose $A^i$ is an irreducible generic, $p$-periodic tensor, and the tensors $A_1, \cdots, A_k$ are obtained from periodic decomposition such that each $A_k$ is normalized and right-canonical. Denote the left dominant eigenvector of $T[\tilde{A}_1]$ as $\tilde{\rho}_1$, whose spectrum is the same as the spectrum of one-cut reduced density matrix of $\left|\psi[\tilde{A}_1]\right\rangle$ due to the injectivity, right-canonicality of $\tilde{A}_1$ and Eq. (9). Denote

$$\tilde{\rho}_{k+1} = (A_1\cdots A_k)^\dagger \tilde{\rho}_1 (A_1\cdots A_k)$$

for $1 \leqslant k \leqslant p-1$. By Lemma 4, $\tilde{\rho}_k$ is the reduced density matrix of $\left|\psi[\tilde{A}_k]\right\rangle$. For brevity we hide the summation of physical indices and use $A^\dagger X A$ to denote the map $\mathcal{E}_{A,l}(X)$.

**Lemma 5.** *The spectrum of $\rho_{o.c.}$ is*

$$\mathrm{eig}(\rho_{o.c.}) = \mathrm{eig}\left(\frac{1}{p}\bigoplus_{k=1}^n \tilde{\rho}_k\right) \qquad (138)$$

*and the following inequality holds,*

$$S(\rho_{o.c.}) \geqslant \ln(2\sqrt{2J+1}), \qquad (139)$$

$$S(\rho_{t.c.}) \geqslant \ln 2(2J+1). \qquad (140)$$

*Proof.* Because the tensor $\tilde{A}^{i_1\cdots i_p}$ is block diagonal and has $p$ blocks, then Eq. (103) implies that the spectrum of $\rho_{o.c.}$ is in the form of Eq. (138).

By the property of entanglement entropy

$$S(\rho_{o.c.}) = \frac{1}{p}(S(\tilde{\rho}_1) + S(\tilde{\rho}_2) + \cdots + S(\tilde{\rho}_p)) + \ln p, \qquad (141)$$

and the inequality

$$S(\tilde{\rho}_k) + S(\tilde{\rho}_{k+1}) \geqslant \ln(2J+1), \qquad (142)$$

we can obtain $S(\rho_{o.c.}) \geqslant \ln(2\sqrt{2J+1})$.

The inequality (142) can be shown as follows. Since the entanglement entropy is determined by the entanglement spectrum, by Proposition 1 we can suppose $\tilde{\rho}_k = \bigoplus_{j_k} t_{j_k} \rho_{j_k}$ in eigenbasis where each $\rho_{j_k}$ is $\frac{1}{2j_k+1} \mathbb{1}_{2j_k+1}$ (here $t_{j_k}$ is determined by the eigenvalues of the matrix $M_{\mu_a^a}^{l,r}$ in Eq. (24)). The direct sum is over all spin sectors with multiplicity counted (i.e., if spin-$j$ has multiplicity $m_j$, then there are correspondingly $m_j$ summands in the direct sum), so the same $j_k$ may appear multiple times in the direct sum. Then

$$\tilde{\rho}_{k+1} = \sum_{j_k} t_{j_k} A_k^\dagger \rho_{j_k} A_k,$$

where we abuse the notation of $\rho_{j_k}$ and its embedding into the full left bond space of $A_k^i$. So the following inequality can be derived,

$$
\begin{aligned}
&S(\tilde{\rho}_k) + S(\tilde{\rho}_{k+1}) \\
=& S\left(\bigoplus t_{j_k} \rho_{j_k}\right) + S\left(\sum_{j_k} t_{j_k} A_k^\dagger \rho_{j_k} A_k\right) \\
\geqslant& \sum_{j_k} \left[ t_{j_k} S(\rho_{j_k}) + (-t_{j_k} \ln t_{j_k}) + t_{j_k} S(A_k^\dagger \rho_{j_k} A_k) \right] \\
\geqslant& \ln(2J+1) + \sum_{t_{j_k}} (-t_{j_k} \ln t_{j_k}) \geqslant \ln(2J+1),
\end{aligned}
$$
(143)

where to obtain the last line we have used

$$
\begin{aligned}
&S(\rho_{j_k}) + S(A_k^\dagger \rho_{j_k} A_k) \\
\geqslant& \min_{|J_1-J_2| \leqslant S \leqslant J_1+J_2} \{ \ln((2J_1+1)(2J_2+1)) \} \\
=& \ln(2J+1).
\end{aligned}
$$
(144)

The inequality (144) comes from the fact that, if there is a spin-$J_1$ sector in the left bond space of $A_k$, there must exist at least one spin-$J_2$ sector in the right bond space of $A_k$ such that $|J_1-J_2| \leqslant S \leqslant J_1+J_2$. Then the eigenspectrum of $A_k^\dagger \rho_{k_k} A_k$ will be a direct sum of some $\rho_{j_{k+1}}$ with $|S-j_k| \leqslant j_{k+1} \leqslant S+j_k$, so $S(A_k^\dagger \rho_{j_k} A_k)$ is greater than or equal to the entanglement entropy of $\rho_{j_{k+1}}$ with smallest $j_{k+1}$ in right bond space of $A_k$.

The equality in Eq. (139) holds only when $p = 2$ and the bond space is composed of $j = 0$ and $j = S$.

In particular, we have shown that

$$S(\rho_{\text{o.c.}}) - \ln(p) \geqslant \ln \sqrt{2J+1} \tag{145}$$

by the above argument. By Eq. (107), we see the corresponding minimal subregion entanglement entropy is $S(\rho_{\text{t.c.}}) = 2(S(\rho_{\text{o.c.}}) - \ln p) + \ln p \geqslant \ln 2(2J+1)$.

Q.E.D.

## B: Minimal entanglement

In the subsection we discuss the minimal entangled states in the set of all non-injective, $SO(3)$-symmetric

uMPS. The result is summarized in Theorem 5. Combined with Theorem 2, we can obtain the lower bound of entanglement entropy of $S(\alpha)$ in the set of *all* uMPS.

**Theorem 5.** *In the set of all non-injective uMPS, the entanglement entropies are lower bounded by*

$$
\begin{aligned}
S(\rho_{\text{t.c.}}) &\geqslant \ln(2(2J+1)), \\
S(\rho_{\text{o.c.}}) &\geqslant \ln(2\sqrt{2J+1}).
\end{aligned}
$$
(146)

*Both lower bounds are saturated by type-II state in Eq.* (20) *with* $\varepsilon = 0$.

In order to prove Theorem 5, we first prove the following lemma, which implies that we only have to prove Theorem 5 for irreducible non-injective uMPS.

**Lemma 6.** *Suppose a symmetric uMPS $A^i$ is a direct sum of some irreducible blocks $A_1^i, \cdots A_n^i$, i.e.,*

$$
A^i = \begin{bmatrix} c_1 A_1^i & & & \\ & c_2 A_2^i & & \\ & & \ddots & \\ & & & c_n A_n^i \end{bmatrix}, \tag{147}
$$

*where each $A_k^i$ is normalized and the weights $c_k$ are positive numbers $c_1 \geqslant c_2 \cdots \geqslant c_n > 0$. Denote $m$ as the number of equally dominant weights,*

$$c_1 = c_2 = \cdots = c_m.$$

*Then the two-cut entanglement entropy $S(\rho_{\text{t.c.}})$ will be no smaller than the following minimum:*

$$\min\{S(\rho_{\text{t.c.}}(A_1^i)), S(\rho_{\text{t.c.}}(A_2^i)), \cdots S(\rho_{\text{t.c.}}(A_m^i))\}$$

*where $\rho_{\text{t.c.}}(A_k^i)$ is the two-cut reduced density matrix from the uMPS $A_k^i$. Similarly, the one-cut entanglement entropy $S(\rho_{\text{o.c.}})$ will be no smaller than the following minimum:*

$$\min\{S(\rho_{\text{o.c.}}(A_1^i)), S(\rho_{\text{o.c.}}(A_2^i)), \cdots S(\rho_{\text{o.c.}}(A_m^i))\}$$

*where $\rho_{\text{o.c.}}(A_k^i)$ is the one-cut reduced density matrix from the uMPS $A_k^i$.*

*Proof.* We present the proof of two-cut entanglement entropy here, and the proof of one-cut entanglement entropy is similar.

First we consider the cases that all $A_k^i$ are inequivalent. Without loss of generality, we can assume that all dominant weights $c_k (1 \leqslant k \leqslant m)$ are equal to 1, so the full tensor $A^i$ is normalized. Because $A_k^i$ are inequivalent, by Lemma 2 in Appendix A, the largest eigenvalues of $T[A]$ must come from the block submatrices $T[A_k]$ with $1 \leqslant k \leqslant m$, and the dominant eigenvectors $T[A_k]$ after being reshaped into matrices are also supported in the subspace corresponding to $A_k^i$. Denote the reduced density matrix from the whole state $|\psi[A]\rangle$

as $\rho_{\text{t.c.}}$, then its expression in Eq. (96) can be simplified to a direct sum of the spectra of each block (see Eq. (99) in Appendix C for details),

$$\text{eig}(\rho_{\text{t.c.}}) = \text{eig}\left(\frac{1}{m}\bigoplus_{k=1}^m \rho_{\text{t.c.}}(A_k^i)\right), \qquad (148)$$

where $\rho_{\text{t.c.}}(A_k^i)$ is the two-cut reduced density matrix from the uMPS $|\psi[A_k]\rangle$ constructed from tensor $A_k^i$.

The entanglement entropy obeys the following equality,

$$S(\frac{1}{m}\bigoplus_k \rho_{\text{t.c.}}(A_k^i)) = \frac{1}{m}\sum_k S(\rho_{\text{t.c.}}(A_k^i)) + \ln m. \qquad (149)$$

Therefore, we conclude that

$$S(\rho_{\text{t.c.}}) \geqslant \min\{S(\rho_{\text{t.c.}}(A_1^i)), S(\rho_{\text{t.c.}}(A_2^i)), \cdots S(\rho_{\text{t.c.}}(A_n^i))\}.$$

Notice that in the above inequality the equality only holds when there is only one irreducible block.

Finally we show that the above statements hold even if some $A_k^i$ are equivalent. In that case, we can put all equivalent $A_k^i$s together as a bigger block, and the reduced density matrix of uMPS of this bigger block is the same as the reduced density matrix of the uMPS of single tensor $A_k^i$ since the two uMPS are the same. Now repeat the argument above we can achieve the conclusion in the lemma.

Q.E.D.

Now we can present the proof of Theorem 5.

*Proof of Theorem 5.* We present the proof of $S(\rho_{\text{t.c.}})$ and the proof of $S(\rho_{\text{o.c.}})$ can be carried out similarly.

By Lemma 6 we only need to consider $A^i$ in the subset of irreducible non-injective tensors. By the discussion above Eq. (131), $A^i$ is $p$-periodic. With the periodic decomposition Eq. (131), we see the dominant eigenvectors $v_k^{l,r}(1 \leqslant k \leqslant p)$ of $T[\tilde{A}]$ will be supported in orthogonal subspaces in the bond space basis, i.e., , in the form of expression (98). Further by Lemma 4 and comments below it, and Lemma 5, we see that the entanglement entropy $S(\rho_{\text{t.c.}})$ is lower bounded by $\ln(2(2J+1))$.

The saturation of the lower bound can be seen through the following computations: the type-II state with $\varepsilon = 0$ has two pairs of dominant eigenvectors:

$$\lambda = \pm 1: \quad v_\lambda^l = \begin{bmatrix} \lambda \\ & \mathbb{1}_{2J+1} \end{bmatrix}, v_\lambda^r = \begin{bmatrix} \lambda \\ & \frac{1}{2J+1}\mathbb{1}_{2J+1} \end{bmatrix}, \qquad (150)$$

put them into Eq. (96) we can get

$$\text{eig}(\rho_{\text{t.c.}}) = \{\frac{1}{2}, \left(\frac{1}{2(2J+1)^2}\right)_{(2J+1)^2}\}$$

where subindex $(2J+1)^2$ is the multiplicity of $\frac{1}{(2J+1)^2}$. Then it is clear $S(\rho_{\text{t.c.}}(\text{II}, \varepsilon = 0)) = \ln(2(2J+1))$.

Q.E.D.

## Appendix I: The minimal Rényi entropy

In this appendix, we first present some details of the proof of Theorem 3, the minimal Rényi entropy in the set of injective uMPS. Then we discuss the minimal Rényi entropy in the set of non-injective uMPS. For $0 < \alpha < 1$, $S_\alpha(\rho)$ is an entanglement monotone which obeys the concavity [48], so the proof for both injective subset and non-injective subset go parallel to the proofs for Theorems 2 and 5. So we mainly consider $\alpha > 1$ in the remaining discussions.

For convenience, we define a function of density matrices

$$H_\alpha(\rho) \equiv \text{Tr}(\rho^\alpha).$$

We list some properties of $H_\alpha(\rho)$:

1. For $\alpha > 1$, $H(\alpha)$ is a convex function in $\rho$, i.e., $H(\sum_i t_i\rho_i) \leqslant \sum_i t_i H(\rho_i)$ for $0 \leqslant t_i \leqslant 1, \sum_i t_i = 1$, and $\rho_i$ some density matrices (see Theorem 3.27 in Ref. [49]).

2. $H_\alpha(\frac{1}{n}\bigoplus_{i=1}^n \rho_i) = \frac{1}{n^\alpha}\sum_i H_\alpha(\rho_i)$

These properties will be used in the following discussions.

Before moving to the proof, we need to understand the relation between one-cut Rényi entropy and two-cut Rényi entropy. If the tensor is injective, then

$$\text{eig}(\rho_{\text{o.c.}}) = \text{eig}(v^l v^r);$$
$$\text{eig}(\rho_{\text{t.c.}}) = \text{eig}((v^l v^r)^{\otimes 2}),$$

so $S_\alpha(\rho_{\text{t.c.}}) = 2S_\alpha(\rho_{\text{o.c.}})$. If the tensor is non-injective and

$$\text{eig}(\rho_{\text{o.c.}}) = \text{eig}\left(\frac{1}{n}\bigoplus_{k=1}^n (v_k^l v_k^r)\right)$$
$$\text{eig}(\rho_{\text{t.c.}}) = \text{eig}\left(\frac{1}{n}\bigoplus_{k=1}^n (v_k^l v_k^r)^{\otimes 2}\right),$$

then

$$S(\rho_{\text{o.c.}}) = -\frac{1}{\alpha - 1}\ln\left(\frac{1}{n^\alpha}\sum_i H_\alpha(\rho_k)\right)$$
$$S(\rho_{\text{t.c.}}) = -\frac{1}{\alpha - 1}\ln\left(\frac{1}{n^\alpha}\sum_i H_\alpha(\rho_k)^2\right), \qquad (151)$$

where $\text{eig}\rho_k = \text{eig}(v_k^l v_k^r)$ For non-injective uMPS, the relation between $S_\alpha(\rho_{\text{o.c.}})$ and $S_\alpha(\rho_{\text{t.c.}})$ is not obvious.

Physically the definition of $S_\alpha(\rho_{\text{t.c.}})$ is more clear so we mainly consider $S_\alpha(\rho_{\text{t.c.}})$. For completeness, we prove the minimality for both quantities.

**Injective uMPS**

We start by proving Theorem 3, the symmetry-enforced minimal Rényi entropy for injective $SO(3)$-symmetric uMPS.

For injective uMPS, the spectra of reduced density matrices $\rho_{\text{o.c.}}$ and $\rho_{\text{t.c.}}$ have been discussed in Sec. III. We only need to consider the set of irreducible injecitive uMPS and search for the minimum of $S_\alpha(\rho_{\text{o.c.}})$. The relation $S_\alpha(\rho_{\text{t.c.}}) = 2S_\alpha(\rho_{\text{o.c.}})$ will ensure that the state with the minimal $S_\alpha(\rho_{\text{o.c.}})$ also has the minimal $S_\alpha(\rho_{\text{t.c.}})$.

First consider the extendable case. The spectrum of the reduced density matrix is in the form of Eq. (27), so by the convexity of $H_\alpha(\rho)$ we can deduce that

$$H_\alpha(\rho_{\text{o.c.}}) \leqslant H_\alpha(\frac{1}{2j_m+1}\mathbb{1}_{2j_m+1}) = \ln(2j_m+1), \quad (152)$$

so $S_\alpha(\rho_{\text{o.c.}}) \geqslant \ln(2j_m + 1)$. For a spin-$J$ system, $j_m \geqslant \frac{J}{2}$, so the minimal one-cut Rényi entropy for extendable uMPS is $S_\alpha(\rho_{\text{o.c.}}) = \ln(J + 1)$. The minimal two-cut Rényi entropy is $S_\alpha(\rho_{\text{t.c.}}) = 2\ln(J+1)$. The minimum is achieved by the type-I state in Eq. (20).

Next, consider the irreducible, generic case (we use "ig" to represent irreducible and generic below). For the ig-injective uMPS, the spectrum of the density matrices is decomposed as shown in Eq. (36). The convexity of $H_\alpha(\rho)$ indicates

$$H_\alpha(X) \leqslant \max\{H_\alpha(\rho^{(1)}), H_\alpha(\rho^{(2)})\},$$

and from the spectrum of $X_1$, we see

$$H(\rho^{(1)}) \leqslant \max\{H(\frac{1}{2}\rho_0 \oplus \rho_J), H(\frac{1}{2}\rho_j \oplus \rho_{f(j)}^{mix})\}$$
$$= H\left(\frac{1}{2}\rho_0 \oplus \rho_J\right) = \frac{1}{2^\alpha}\left(1 + \frac{1}{(2J+1)^{\alpha-1}}\right), \quad (153)$$

where the inequality is proved later in the Eq. (166). The other candidate $H_\alpha(\rho^{(2)})$ is already discussed in the extendable case. For the two-cut entropy $S_\alpha(\rho_{\text{t.c.}})$, since the tensor is ig-injective, $S_\alpha(\rho_{\text{t.c.}}) = 2S_\alpha(\rho_{\text{o.c.}})$. This concludes the proof of Theorem 3.

**Non-injective uMPS**

Next, we turn to non-injective uMPS. We first present the theorem of the symmetry-enforced lower bound of Rényi-$\alpha$ entropy in the set of all non-injective uMPS.

**Theorem 6.** *In the set of non-injective, symmetric uMPS of spin-J ($J \in \mathbb{Z}^+$) chain, the Rényi-$\alpha$ entropies*

*are bounded by*

$$S_\alpha(\rho_{t.c.}) \geqslant -\frac{1}{\alpha-1}\ln\left(\frac{1}{2^\alpha}(1 + \frac{1}{(2J+1)^{2\alpha-2}})\right)$$
$$S_\alpha(\rho_{o.c.}) \geqslant -\frac{1}{\alpha-1}\ln\left(\frac{1}{2^\alpha}(1 + \frac{1}{(2J+1)^{\alpha-1}})\right)$$
$$(154)$$

*Both lower bounds are tight bounds, and are saturated by the type-II states with $\varepsilon \to 0$ in Eq. (20).*

The proof strategy is parallel to the proof of Theorem 5. First, we do reduction to restrict to the irreducible tensors. Following the setup of Lemma 6, we want to show

$$S_\alpha(\rho_{\text{t.c.}}) \geqslant \min\{S_\alpha(\rho_{\text{t.c.}}(A_1^i)), S_\alpha(\rho_{\text{t.c.}}(A_2^i)), \cdots, S_\alpha(\rho_{\text{t.c.}}(A_n^i))\}.$$
$$(155)$$

$$S_\alpha(\rho_{\text{o.c.}}) \geqslant \min\{S_\alpha(\rho_{\text{o.c.}}(A_1^i)), S_\alpha(\rho_{\text{o.c.}}(A_2^i)), \cdots, S_\alpha(\rho_{\text{o.c.}}(A_n^i))\}.$$
$$(156)$$

From Eq. (148) and the convexity of $H_\alpha(\rho)$, we see

$$H_\alpha(\rho_{\text{t.c.}}) \leqslant \frac{1}{m}\sum_k H_\alpha(\rho_{\text{t.c.}}(A_k^i)) \leqslant \max_k\{H(\rho_{\text{t.c.}}(A_k^i))\},$$
$$(157)$$

so applying $-\frac{1}{\alpha-1}\ln(\cdot)$ on two sides we obtain inequality (155). Similarly one can show the statement for $\rho_{\text{o.c.}}$.

Next, to deal with the $p$-periodic cases we need an analog of Lemma 5, which we put as Lemma 7. With this lemma we can directly follow the proof of Theorem 5 in Sec. H B and find the minimum of $S_\alpha(\rho_{\text{o.c.}})$ and $S_\alpha(\rho_{\text{t.c.}})$.

**Lemma 7.** *Consider the setup of Lemma 5. Then the spectrum of the reduced density matrix is*

$$\text{eig}(\rho_{t.c.}) = \frac{1}{p}\left(\bigoplus_{k=1}^p \tilde{\rho}_k^{\otimes 2}\right)$$

*and the following inequality holds,*

$$H_\alpha(\rho_{o.c.}) \leqslant \frac{1}{2^\alpha}\left(1 + \frac{1}{(2J+1)^{\alpha-1}}\right)$$
$$H_\alpha(\rho_{t.c.}) \leqslant \frac{1}{2^\alpha}\left(1 + \frac{1}{(2J+1)^{2\alpha-2}}\right)$$
$$(158)$$

*Proof.* The proof goes similarly as the proof of Lemma 5. Denote $\rho_{k+1} = A_k^\dagger \rho_k A_k$ for $1 \leqslant k < p$

$$H_\alpha(\rho_{\text{o.c.}}) = \frac{1}{p^\alpha}\sum_{k=1}^p H_\alpha(\tilde{\rho}_k) \quad (159)$$

Using the inequality

$$H_\alpha(\tilde{\rho}_k) + H_\alpha(\tilde{\rho}_{k+1}) \leqslant \max_j\{H_\alpha(\rho_j) + H_\alpha(\rho_{S-j})\}$$
$$= 1 + \frac{1}{(2J+1)^{\alpha-1}},$$
$$(160)$$

we find

$$H_\alpha(\rho_{\text{o.c.}}) \leqslant \frac{1}{2^\alpha}\left(1 + \frac{1}{(2J+1)^{\alpha-1}}\right) \qquad (161)$$

For two-cut entropy, from Eq. (151) $H_\alpha(\rho_{\text{t.c.}})$ can be expressed as

$$H_\alpha(\rho_{\text{t.c.}}) = \frac{1}{p^\alpha}\sum_{k=1}^{p} H_\alpha(\rho_k)^2. \qquad (162)$$

A similar argument to inequality (160) shows

$$H_\alpha(\rho_k)^2 + H_\alpha(\rho_{k+1})^2 \leqslant \max_j\{H_\alpha(\rho_j)^2 + H_\alpha(\rho_{S-j})^2\}$$
$$= 1 + \frac{1}{(2J+1)^{2\alpha-2}}, \qquad (163)$$

so finally we see

$$H_\alpha(\rho_{\text{t.c.}}) \leqslant \frac{1}{2^\alpha}\left(1 + \frac{1}{(2J+1)^{2\alpha-2}}\right). \qquad (164)$$

Q.E.D.

**Proof of the inequality** (153)

Finally, for completeness, we prove the inequality (153).

In general,

$$X_1 \propto \bigoplus_{j<\frac{J}{2}} t_{1,j}\rho_j, \quad \rho_j = \frac{1}{2j+1}\mathbb{1}_{2j+1}, \sum_j t_{1,j} = 1$$

so from $\rho^{(1)} \propto X_1 \oplus B_1^\dagger X_1 B_1$

$$\rho^{(1)} = \sum_{j<\frac{J}{2}} \frac{t_{1,j}}{2}(\rho_j \oplus \mathcal{E}_{A_1,l}(\rho_j)). \qquad (165)$$

With the help of Jensen's inequality, and the diagonal nature of every $\rho_j$, we see

$$H_\alpha(\rho^{(\alpha)}) \leqslant \sum_{j<\frac{J}{2}} t_{1,j}\left(\frac{1}{2^n}\text{Tr}[\rho_j^n \oplus \mathcal{E}_{A_1,l}(\rho_j)^n]\right)$$
$$\leqslant \max_{j<\frac{J}{2}}\left\{\left(\frac{1}{2^n}\text{Tr}[\rho_j^n \oplus \mathcal{E}_{A_1,l}(\rho_j)^n]\right)\right\} \qquad (166)$$
$$\leqslant \max_{j<\frac{J}{2}}\left\{\left(\frac{1}{2^n}\text{Tr}[\rho_j^n \oplus \rho_{S-j}^n]\right)\right\}$$
$$= \frac{1}{2^n}(1 + \frac{1}{(2J+1)^{n-1}})$$

where the last line is true for $j = 0, S - j = S$, i.e., the valence bond case. The last line is obtained by the monotonicity of the second last line with respect to $j$.

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
