# Peer review of "Symmetry-enforced minimal entanglement and correlation in quantum spin chains"

_SciPost Physics_

## Round 1 · Referee Report · Anonymous (Referee 1) · 2025-2-11

Strengths

1-A clear and rigorous analysis of the minimal entanglement and correlation enforced by symmetries in quantum spin-J chains.
2-Well organized structure of major conclusions and proof.

Weaknesses

1-Lack of discussion of physical insights and applications.

Report

This manuscript investigates the minimal entanglement and correlation for one-dimensional spin chains enforced by translational and SO(3) symmetries. Utilizing the translational symmetries, it provides a rigorous mathematical proof based on symmetric uniform MPS. The manuscript is well-structured, presenting clear and solid proofs. Overall, it meets the criteria for publication with minor improvement.

Requested changes

1-Although a rigorous proof is provided, there is lack of elaboration of intuitive understanding and physical insights. What are the insights and potential applications of your results?
2-What are the expectations of the minimal entanglement and correlation for more general settings, such as those that are also mentioned in the discussion section, e.g., for other symmetries and higher spatial dimensions? Are there obstruction to the generalizations?

Recommendation

Publish (easily meets expectations and criteria for this Journal; among top 50%)

  • validity: high
  • significance: good
  • originality: high
  • clarity: high
  • formatting: good
  • grammar: good

Author:  Liujun Zou  on 2025-03-19  [id 5299]

(in reply to Report 1 on 2025-02-11)

The referee writes:

"This manuscript investigates the minimal entanglement and correlation for one-dimensional spin chains enforced by translational and SO(3) symmetries. Utilizing the translational symmetries, it provides a rigorous mathematical proof based on symmetric uniform MPS. The manuscript is well-structured, presenting clear and solid proofs. Overall, it meets the criteria for publication with minor improvement."

Our response:

We thank the referee for the nice comments and suggestions on the manuscript, and for the recognition of this work as worthy of publication.

The referee writes:

"1-Although a rigorous proof is provided, there is lack of elaboration of intuitive understanding and physical insights. What are the insights and potential applications of your results?"

Our response:

As discussed in the Introduction, the combination of semiclassical consideration and the Lieb-Schultz-Mattis theorem suggests that the minimal entanglement entropy should diverge as $J$ increases. This intuition is verified by our rigorous calculations. However, this intuition does not tell us which states are the minimally entangled states and what the values of the minimal entanglement entropies are.

An intuitive understanding of why the AKLT-like states can be the minimally entangled states is from the perspective of the bond space dimension counting. Very roughly speaking, a larger bond dimension typically yields larger entanglement. For a spin-$J$ chain, the smallest bond dimension is $J+1$, corresponding to the AKLT-like state with a single spin-$\frac{J}{2}$ sector in the bond Hilbert space. So one may expect that the AKLT-like states are the minimally entangled states. However, this expectation is correct only for some values of $J$, while for other values of $J$ the minimally entangled states are the type-II states. We need to carry out the detailed analysis to determine the minimally entangled states.

A potential application of our results is to use the AKLT-like states as an initial seed to find the minimal correlation length in symmetric states. As discussed in our paper, the minimally entangled states do not exhibit the smallest correlation length. But due to their simplicity in the tensor language, these AKLT-like states serve as a good starting point to search the state with the smallest correlation length. For instance, we can add the off-diagonal spin blocks into the tensor of the AKLT-like states and see how the correlation length varies. This provides a new and interesting perspective to study the relation between correlation length and entanglement measures.

We have added these comments into the Discussion section.

The referee writes:

"2-What are the expectations of the minimal entanglement and correlation for more general settings, such as those that are also mentioned in the discussion section, e.g., for other symmetries and higher spatial dimensions? Are there obstruction to the generalizations?"

Our response:

The generalizations to more general settings are expected as follows. For other symmetries which are non-Abelian and continuous, such as $SO(N),SU(N)$, we expect that similar conclusions of the minimal entanglement can be obtained. This is because the symmetric uMPS structure is valid for general symmetries, and our Proposition 1 can be generalized to these symmetries. We also expect that a zero correlation length is impossible in the presence of these symmetries. However, we anticipate some fine differences between the case with these other symmetry groups and the case with $SO(3)$, because some very specific properties of the $SO(3)$ group are crucial in some parts of our derivation of the minimal entanglement.

For higher spatial dimensions the problem is more complicated. First, in higher dimensions the geometry of the subsystem does influence the value of the entanglement measures such as the von Neumann entropy. Therefore, it requires some care to even define the notion of the minimal entanglement. Second, the symmetric tensor network description for higher dimensions is more complicated, and it is not clear to us how to obtain the structure of entanglement or reduced density matrices of Projected-Entangled-Pair-States (PEPS) yet.

---

## Round 1 · Referee Report · Anonymous (Referee 2) · 2025-3-14

Strengths

1- Detailed exposition of uniform MPS, transfer matrix, entanglement entropies and symmetries. 2- A derivation of the minimal entanglement entropy for symmetric MPS (translation symmetry + internal SO(3) symmetry)

Weaknesses

1- The central result on the minimal entanglement is well-known from the perspective of symmetric MPS 2- The observation that minimal correlation length and minimal entanglement do not coincide, is left unproven

Report

In this work, the authors consider the general form of MPS in the presence of translation symmetry and internal SO(3) symmetry. From this general form, they derive that there are lower bounds on the entanglement entropy in the MPS. On top of that, it is shown that saturating this lower bound does not mean necessarily that the correlation length is minimal.

The topic of symmetric MPS is important on the analytical side (where it has led to the classification of all SPT phases in one dimension) and the numerical side (where it leads to the most efficient algorithms for simulating quantum spin chains with symmetries). The case of SO(3) symmetry is paradigmatic in that sense, as this includes the first example of an SPT phase (the Haldane phase in the spin-1 Heisenberg chain, with the AKLT state as the minimal MPS description) and one of the primordial examples of a system that can be treated with MPS techniques. For that reason, the MPS symmetry structure is well known in that case.

If I understand correctly, the central result in the paper is the fact that generic MPS in an spin-J chain with minimal entanglement entropy either have a J/2-representation on the bond, or a (0+J) representation. In my opinion, this is an almost trivial result: it is well known that MPS with integer-J spin chains allow for SPT-like order (half integer bond representations) or trivial order (integer bond representations). Given that fact, it is quite clear that the minimal representations are the ones presented in the paper. The explicit MPS representations for these two choices are also quite straightforward generalizations of the AKLT construction.

That these minimal-entanglement MPS (i.e. the AKLT-like states) do not exhibit minimal correlation lengths is also not entirely unexpected, as there is no good reason why they should. Since the authors are mainly presenting this as an observation, and corroborate this with a very simple numerical experiment, the reader does not gain any new real insight here as well.

In conclusion, I do not think the results in this paper are original and of high relevance to the research field. Nonetheless, the paper is well-written, contains some interesting observations and explicit proofs, so I think the paper can be interesting for many readers. Therefore, I would suggest the paper to be transferred to Scipost Physics Core.

Recommendation

Accept in alternative Journal (see Report)

  • validity: high
  • significance: low
  • originality: low
  • clarity: high
  • formatting: perfect
  • grammar: perfect

Author:  Liujun Zou  on 2025-03-19  [id 5300]

(in reply to Report 2 on 2025-03-14)

The referee writes:

"In this work, the authors consider the general form of MPS in the presence of translation symmetry and internal SO(3) symmetry. From this general form, they derive that there are lower bounds on the entanglement entropy in the MPS. On top of that, it is shown that saturating this lower bound does not mean necessarily that the correlation length is minimal.

The topic of symmetric MPS is important on the analytical side (where it has led to the classification of all SPT phases in one dimension) and the numerical side (where it leads to the most efficient algorithms for simulating quantum spin chains with symmetries). The case of SO(3) symmetry is paradigmatic in that sense, as this includes the first example of an SPT phase (the Haldane phase in the spin-1 Heisenberg chain, with the AKLT state as the minimal MPS description) and one of the primordial examples of a system that can be treated with MPS techniques. For that reason, the MPS symmetry structure is well known in that case.

If I understand correctly, the central result in the paper is the fact that generic MPS in an spin-J chain with minimal entanglement entropy either have a J/2-representation on the bond, or a (0+J) representation. In my opinion, this is an almost trivial result: it is well known that MPS with integer-J spin chains allow for SPT-like order (half integer bond representations) or trivial order (integer bond representations). Given that fact, it is quite clear that the minimal representations are the ones presented in the paper. The explicit MPS representations for these two choices are also quite straightforward generalizations of the AKLT construction."

Our response:

We thank the referee for pointing out an interesting perspective on our results. We would like to respond from the following three aspects:

First, the referee seems to miss one of the main results of our work. In addition to the minimal entanglement and the inequivalence between minimal entanglement and minimal correlation length, we also show that the correlation length cannot be zero. This point does not appear in the summary of our paper by the referee.

Second, the referee's main criticism of the minimal entanglement part is that the result is intuitively expected. However, even if a result is intuitively expected, its rigorous derivation is still valuable if the derivation is nontrivial. Consider another example, the entanglement area law of a ground state of a gapped local Hamiltonian is also intuitively expected, but its proof is still very important.

Third, the intuition of the referee is that "it is well known that MPS with integer-J spin chains allow for SPT-like order (half integer bond representations) or trivial order (integer bond representations). Given that fact, it is quite clear that the minimal representations are the ones presented in the paper." This intuition is actually incorrect. As we remark in the paper, the (0, J) representation has neither the SPT order nor the trivial order. Instead, it has a spontaneous symmetry breaking (SSB) order, which is beyond the above intuitive expectation (note that the states which do not spontaneously break the symmetry can have entanglement arbitrarily close to this state, as we discuss in the paper). Moreover, even if one has SPT order, trivial order and SSB order all in mind, it is not obvious which state has the minimal entanglement, since for each order there are infinitely many states and careful analysis is needed. Additionally, specific properties of the SO(3) group are crucial in our rigorous derivation of the minimal entanglement, and it is not entirely straightforward to generalize our results to other internal symmetry groups. Therefore, any intuition based on the fact that there are SPT order, trivial order and SSB order is unlikely to hold in general.

The referee writes:

"That these minimal-entanglement MPS (i.e. the AKLT-like states) do not exhibit minimal correlation lengths is also not entirely unexpected, as there is no good reason why they should. Since the authors are mainly presenting this as an observation, and corroborate this with a very simple numerical experiment, the reader does not gain any new real insight here as well."

Our response:

We agree that we did not fully solve the problem of the minimal correlation length, which is also explicitly commented on in the draft. However, we believe that our discussion is still physically meaningful for the following reasons.

It is true that we do not have to expect that the minimally entangled states exhibit the minimal correlation length. Nevertheless, the minimally entangled states (in particular, the AKLT-like states) provide a natural starting point to study the minimal correlation length, since it has a relatively simple tensor structure. Its correlation length can be regarded as a reference, and we can ask how the correlation length of the state changes when the tensor of the AKLT-like state is perturbed. This may be an interesting insight for studying the relation between correlation length and entanglement measures.

As for the current manuscript, we remark that the statement "the minimally entangled MPS do not exhibit the minimal correlation lengths'' is just a feature of our first-step exploration, not the most essential part of the paper. We keep the record of some numerical results to show that we can lower the correlation length by adding off-diagonal spin blocks in the tensor, which has not been explicitly discussed previously, as far as we know.

On the other hand, the data of the small correlation length we found can be interesting on its own. As mentioned, the AKLT-like states provide a good reference scale of correlation length to start with, and value we found is the smallest that we can find through global minimization using a laptop. For future research, our data can be an interesting object to compare with.

The referee writes:

"In conclusion, I do not think the results in this paper are original and of high relevance to the research field. Nonetheless, the paper is well-written, contains some interesting observations and explicit proofs, so I think the paper can be interesting for many readers. Therefore, I would suggest the paper to be transferred to Scipost Physics Core."

Our response:

Summarizing the replies above, we believe that our paper is valuable and suitable for being published on SciPost Physics.

---

## Editorial Decision

resubmitted